# Beyond the facility: An evaluation of seven community-based pediatric HIV testing strategies and linkage to care outcomes in a high prevalence, resource-limited setting

Kathleen Sindelar[1]*, Chivimbiso Maponga[1], Fumane Lekoala[1], Esther Mandara[1], Matsitso Mohoanyane[2], Jill Sanders[3], Jessica Joseph[4]*

**1** Clinton Health Access Initiative, Maseru, Lesotho, **2** Lesotho Ministry of Health, Maseru, Lesotho, **3** Baylor College of Medicine Children's Foundation – Lesotho, Maseru, Lesotho, **4** Clinton Health Access Initiative, Boston, Massachusetts, United States of America

* kjsindelar@gmail.com (KS); jjoseph@clintonhealthaccess.org (JJ)

**Data Availability Statement:** Anonymized data derived directly from the Mobilizing HIV

## Abstract

Diverse challenges in expanding pediatric HIV testing and treatment coverage persist, making the investigation and adoption of innovative strategies urgent. Evidence is mounting for the effectiveness of community-based testing in bringing such lifesaving services to those in need, particularly in resource-limited settings. The Mobilizing HIV Identification and Treatment project piloted seven community-based testing strategies to assess their effectiveness in reaching HIV-positive children and linking them to care in two districts of Lesotho from October 2015 to March 2018. Children testing HIV-positive were enrolled into the project's mHealth system where they received e-vouchers for transportation assistance to the facility for treatment initiation and were followed-up for a minimum of three months. An average of 7,351 HIV tests were conducted per month across all strategies for all age groups, with 46% of these tests on children 0–14 years. An average of 141.65 individuals tested positive each month; 9% were children. Among the children tested 55% were over 5 years. The yield in children was low (0.38%), however facility-based yields were only slightly higher (0.72%). Seventy-five percent of children were first-time testers and 86% of those testing HIV-positive were first-time testers. Seventy-one percent of enrolled children linked to care, all but one initiated treatment, and 82% were retained in care at three months. As facility-based testing remains the core of HIV programs, this evaluation demonstrates the effectiveness of community-based strategies in finding previously untested children and those over 5 years who have limited interactions with the conventional health system. Utilizing active follow-up mechanisms, linkage rates were high suggesting accessing treatment in a facility after community testing is not a barrier. Overall, these community-based testing strategies contributed markedly to the HIV testing landscape in which they were implemented, demonstrating their potential to help close the gap of unidentified HIV-positive children and achieve universal testing coverage.

Identification and Treatment project have been uploaded as Supporting Information files with this submission. However, one dataset presented with this research that was extracted from Lesotho's Ministry of Health data warehouse, DHIS2, is unavailable as we do not possess the rights to legally distribute it. This dataset contains all the numbers on facility-based HIV testing and positive identifications that are provided in this manuscript to contextualize the project's findings. Permission to use this dataset for analysis and share the results in a peer reviewed journal was obtained from the Ministry of Health's Ethical Committee during the project's initial research protocol application (ID68-2015). Requests for this data may be directed to Lesotho's Ministry of Health Research Coordination Unit. The contact details for this unit may be obtained from Mr. Tumisang Mokoai at +266 58850055 or kkpromos@gmail.com.

**Funding:** This project was made possible through a consortium of funders that was headed by the Vodafone Foundation. Other funders include the U. S. Agency for International Development (USAID), the ELMA Foundation, Elton John AIDS Foundation, and Viiv Healthcare. No grant numbers were assigned for this project. None of the aforementioned funders played a role in the design of the study, metrics measured, data collection and analysis, decision to publish, or the preparation of the manuscript.

**Competing interests:** This project was made possible through a consortium of funders that was headed by the Vodafone Foundation. Other funders include the U.S. Agency for International Development (USAID), the ELMA Foundation, Elton John AIDS Foundation, and Viiv Healthcare. No grant numbers were assigned for this project. Vodafone Foundation is a UK registered charity (1089625), which is funded by annual contributions from Vodafone Group Plc. Vodafone Group partially owns Vodacom Group, the parent company of Vodacom Lesotho, who is the operator of m-pesa, the mobile money system utilized in this research and discussed in this paper. None of the aforementioned funders played a role in the design of the study, metrics measured, data collection and analysis, decision to publish, or the preparation of the manuscript. No incentives, financial or otherwise, were received by Vodafone Foundation or any implementing partner for the use of m-pesa within the Mobilizing HIV Identification and Treatment project. This does not alter our adherence to PLOS ONE policies on sharing data and materials.

## Introduction

Substantial gains have been achieved in expanding HIV testing and antiretroviral therapy (ART) access for children 0–14 years worldwide. Global estimates indicate an increase in ART coverage from 127,300 in 2006 to 940,000 in 2017 [1]. However, coverage remains disproportionately low with only 52% [33–70%] of children accessing ART compared to 59% [44–73%] of adults [2]. Common systemic and provider challenges within pediatric testing and treatment include lack of programs that integrate the specific needs of children [2–4], lack of personnel who are sufficiently trained in providing HIV services for children [1, 4], poor linkage and retention mechanisms [3, 5–7], challenges in early infant diagnosis (EID) [1–3, 5, 8, 9], as well as frequent stockouts of pediatric antiretroviral formulations and other essential commodities [4, 6, 10]. Challenges also occur at the household level due to parental consent laws coupled with stigma [6, 11] and economic constraints that prevent children from accessing healthcare services [12]. These challenges contribute to delayed testing that often only occurs once children have become symptomatic and reached advanced stages of the disease, resulting in unnecessary morbidity and mortality [3, 4, 11, 13–16].

Acknowledging that HIV testing services (HTS) are the first step toward increasing treatment coverage and saving lives, innovative strategies must be explored and evaluated [11, 13]. One proven approach to expanding access to HTS is through community-based testing (CBT) [17–23], a broad term that refers to services provided outside of clinical settings [19]. CBT has been shown to overcome many common barriers to care, allowing individuals to be detected earlier in the course of infection and reducing the burden on the health system [2, 11, 20, 24–27]. Offering parallel facility and community-based HTS has been shown to increase awareness of HIV status at the population level 7-fold [28]. However, little research exists that focuses on the impact of CBT among children.

The adoption of CBT is essential in a country such as Lesotho that has a severe generalized epidemic with a 25% prevalence rate [29], coupled with many remote communities who face geographic and economic challenges in accessing the formal health system [24]. In 2015, all health facilities in Lesotho offered provider-initiated and voluntary HTS, yet the country was believed to be far from achieving the 90-90-90 targets (90% of HIV-positive people knowing their status, 90% of HIV-positive people who are aware of their status on ART, and 90% of people on ART virally suppressed). National estimates indicated that only 5,700 HIV-positive children of the estimated 19,000 (33%) were receiving ART [30] and Prevention of Mother-to-Child Transmission (PMTCT) coverage was only 63.9% [31]. The limited HTS opportunities available outside of facilities failed to meet the needs of the growing and hard-to-reach population of children who slipped through the PMTCT cascade.

As a result, the Ministry of Health (MoH), donors, and implementing partners identified CBT as a tool to close the gap of unidentified HIV-positive children. In October 2015, the Mobilizing HIV Identification and Treatment (M-HIT) project commenced with a focus on rapidly identifying HIV-positive children, as well as pregnant and lactating women, and linking them to treatment through seven different community-based testing strategies in two of Lesotho's largest and most heavily burdened districts. Active client follow-up was an essential component of the program to account for high attrition rates documented among many CBT studies in the region [26]. Four e-vouchers were automatically sent to newly identified HIV-positive patients to assist in overcoming the commonly cited barrier of transport costs [32].

This paper examines the outcomes of M-HIT's seven CBT strategies and assesses their effectiveness at reaching children for HTS, as well as identifying HIV-positive children and linking them to care and treatment. HTS uptake was not captured across all strategies and will not be presented in this paper.

**Abbreviations: ANC**, Antenatal care; **ART**, Antiretroviral therapy; **BCMCF-L**, Baylor College of Medicine Children's Foundation–Lesotho; **CBT**, Community-Based Testing; **CHAI**, Clinton Health Access Initiative; **D2D**, Door-to-door testing; **DNA PCR**, Deoxyribonucleic acid polymerase chain reaction; **EID**, Early Infant Diagnosis; **HTS**, HIV testing services; **LSL**, Lesotho Loti; **M-HIT**, Mobilizing HIV Identification and Treatment; **MOC**, Mobile outreach clinic; **MoH**, Ministry of Health; **PMTCT**, Prevention of Mother-to-Child Transmission; **PSI**, Population Services International; **R4H**, Riders for Health; **USD**, United States Dollar.

## Objectives

The primary objectives were as follows:

1. Evaluate HTS delivered through each community-based testing strategy by measuring the proportion of children (0–14 years) tested, the number of children tested, the number of children identified as HIV-positive, and the percentage testing positive (yield).

2. Compare the number of children (0–14 years) identified as HIV-positive in each M-HIT strategy to the number of children identified as HIV-positive in standard practice (i.e. facility-based testing).

3. Determine if children testing HIV-positive in a community-based setting successfully linked to care and initiated ART within three months.

## Methods

### Study design

M-HIT was a non-experimental prospective project conducted in two districts in Lesotho from October 2015 to March 2018. The project implemented and evaluated seven CBT strategies that aimed to identify, enroll, and link children aged 0 to 14 years who tested HIV-positive and were not already on ART. All enrolled children were then followed-up for a minimum of three months to track linkage to care, treatment, and retention outcomes.

### Study population and sampling method

The districts of Maseru and Leribe were selected as they had the highest estimated volume of unidentified HIV-positive children at the start of the project. The testing strategies covered the catchment areas of all government funded health facilities. The overall evaluation population included all individuals residing or seeking care in both districts. No sampling procedure was implemented as all individuals who received M-HIT services were included in the analysis to maximize statistical power to detect effects.

### Intervention: M-HIT project overview

**Community-based testing strategies.** M-HIT was a collaborative effort among five partners in Lesotho with two components: testing, and linkage to care with active follow-up for those identified as HIV-positive. The project was designed to be implementation research with testing and follow-up activities conducted by implementing partners, but also integrated into the national health system. M-HIT's evaluation was conducted for a fixed time period to monitor activities and determine effectiveness. Two implementing partners conducted and supported the testing strategies and linkage to care with follow-up: Baylor College of Medicine Children's Foundation–Lesotho (BCMCF-L) and Population Services International (PSI). HTS was provided by nurses and counselors trained in pediatric HTS. Riders for Health (R4H) provided logistical support for the mobile outreach clinics, and Clinton Health Access Initiative (CHAI) conducted program planning, management, linkage to care documentation, and the evaluation activities. Finally, the Lesotho MoH–at central, district, and health facility level–were key implementing partners, advisors, and stakeholders.

The seven M-HIT strategies are described in Table 1. Every strategy focused on providing HTS for children 0–14 years and pregnant and lactating women at a community level; however, HTS were offered to individuals of all ages in all strategies. Testing strategies were either

**Table 1. Description of community-based testing strategies.**

| Testing Strategy | Description | Implementing Partners | Implementation Period | Strategy Type |
|---|---|---|---|---|
| Mobile Outreach Clinics (MOC) | Comprehensive health services (e.g. HTS, antenatal care (ANC), immunization, outpatient) provided in a rural community through their respective government funded health facility on a monthly basis | BCMCF-L; MoH | Oct 2015-Mar 2018 (30 months) | Blanket |
| Targeted Testing | Provision of HTS at venues with children believed to be at high-risk of HIV infection due to known high prevalence among parents (e.g. orphanages, informal daycare centers for the children of factory workers) | PSI | Nov 2015-Apr 2016, Sep 2016-Oct 2016 (8 months) | Targeted |
| Facility Index Testing | Home-based HTS offered to all household members of ART clients who gave consent at facilities during drug refill visits | PSI | Oct 2015-Mar 2016, Jun 2016-Jul 2016, Oct 2016-Nov 2016, Jun 2017-Jul 2017 (12 months) | Targeted |
| Door-to-Door Testing (D2D) | Houses that surround the residence of facility-index testing clients were approached and offered HTS to conceal the HIV-positive status of facility-indexed client | PSI | Oct 2015-Mar 2018 (30 months) | Blanket |
| Door-to-Door Index Testing | Occurred during D2D testing when an individual of a D2D household self-reported as HIV-positive—this person became an index client and all other household members were immediately offered HTS | PSI | Oct 2015-Mar 2018 (30 months) | Targeted |
| Roadside Tent Testing | Tents erected for several days in urban and semi-urban high-traffic pedestrian locations to provide HTS to eligible individuals | PSI | Feb 2016–Mar 2018 (26 months) | Blanket |
| Semi-Static Roadside Tent Testing | Tents erected in urban and semi-urban high-traffic pedestrian locations for a minimum of one month to provide HTS to eligible individuals | PSI | Jan 2017, Apr 2017-Nov 2017 (9 months) | Blanket |

targeted—specific attempts to find populations of children with higher HIV prevalence, or blanket—providing HTS to all children regardless of risk factors.

## Testing and follow-up procedures

HTS eligibility for all age cohorts was based on the MoH guidelines [33]. Eligible individuals included anyone with an unknown status, >18 months. Eligibility also included all adolescents (in this case the MoH defined this cohort as those 10–19 years) and adults who reported being sexually active that had not been tested in the past year. Additionally, it included those who reported recent or ongoing risk of exposure who had not been tested in the past three months.

Within the M-HIT project, both index testing strategies were the exception to the afore-mentioned HTS eligibility requirements as they tested all household members regardless of testing history. As stated in Table 1 in the description for Facility Index testing, all household members were offered HTS once consent from the ART patient was received. Similarly, during D2D Index testing, all household members were immediately offered HTS after someone self-reported as HIV-positive. The reason for expanding HTS eligibility in this context is due to the belief that these individuals were at higher risk of infection.

Eligibility for infants (0–18 months) was based on the HIV status of the mother. If the mother was not present and her status was unknown, the infant was administered a rapid test to determine exposure status. Infants testing positive were referred to a health facility to receive confirmatory DNA PCR testing and were captured in the project's mHealth system for client follow-up. For HIV-exposed infants, EID algorithms were followed, although DNA PCR testing was not available for all strategies and is therefore not reported in this paper. During follow-up it was discovered that some children had a previous HIV-positive diagnosis, but had not yet initiated on treatment; therefore, they remained in the follow-up cohort.

Procedures for consent were also consistent with MoH guidelines and was obtained by implementing partners at the time of testing [33]. This required verbal consent for anyone 12 years and older (the legal age of consent in Lesotho) and written consent from the parent or guardian present during HTS for children under 12 years.

Prior to Lesotho's adoption of the Treat All policy (all HIV-positive individuals to begin treatment immediately after diagnosis), children who tested positive at an M-HIT testing event were referred to their preferred health facility for CD4 testing or treatment initiation. Per national treatment guidelines, all HIV-positive children under 5 years were eligible for ART initiation, as were those over 5 years with a CD4 count <500 and those coinfected with tuberculosis or hepatitis B [33]. After the national implementation of Treat All in June 2016 some MOCs were able to offer immediate ART initiation to children on-site. Children were to return to the facility two weeks after initiating ART and subsequently every month to receive drug refills.

Initial client follow-up was conducted by lay counselors and commenced when a newly identified HIV-positive client did not present at the facility within two weeks of a positive test. Counselors attempted to contact the parent or guardian through the mobile phone number provided. If they were not reached within two weeks or refused treatment for their child, a counselor or village health worker would visit their home for further HIV education and counseling.

## mHealth enrollment and mobile money system (M-pesa)

The project utilized a customized mHealth system to enroll the newly identified HIV-positive children at the point-of-diagnosis into a follow-up cohort. Verbal consent for enrollment was required from all clients or the parent or guardian of children under 12 years of age prior to enrollment. M-Pesa, a mobile money system operated by Vodacom Lesotho, was utilized to distribute four unconditional e-vouchers to enrolled clients to serve as reimbursement for their transportation costs from their home to a preferred health facility for treatment. These e-vouchers ranged between 12 LSL—50 LSL each (~USD$1–$5) and their electronic delivery aligned with the client's ART initiation appointments. Due to delays in the development of the mHealth system, electronic enrollment began in December 2015 and concluded in November 2017, allowing for a further three months of follow-up data collection. Therefore, all follow-up cohort analysis only includes the HIV-positive children identified in this time period.

## Data collection

Monthly service delivery data on the number of rapid tests conducted and the corresponding results, disaggregated by age and sex when available, as well as other key indicators were collected from implementing partners for all strategies. BCMCF-L submitted data on MOCs in aggregate while PSI provided individual-level data for all other testing strategies. Lesotho's Ministry of Health data warehouse, DHIS2, provided aggregated testing numbers and results for all facility-based testing that occurred in the same period.

General client information and testing history was collected by project healthcare workers into the mHealth system on all HIV-positive children at the time of testing. Subsequently, data collectors routinely visited health facilities to document facility attendance, ART initiation, and refill dates from facility ART registers.

Linkage to care and ART initiation were measured as a single documented pre-ART or ART initiation visit at an MOC or health facility within three months of the HIV-positive test date. Similarly, retention in care was also documented by ART refill visits and was defined as being engaged in care by a refill date at least three months after initiation.

Ethics approval was obtained from Lesotho's MoH Research and Ethics Committee (ID68-2015), as well as Chesapeake Institutional Review Board (MOD00240045) for the duration of the M-HIT project and evaluation. Informed consent for the evaluation was waived by both IRBs as consent for services was already received by the implementing partners during the testing and follow-up process. While identifiable information was collected on patients in order to conduct follow-up on linkage to care, once completed the dataset was de-identified for analysis.

## Data analysis

The number tested and the number identified as HIV-positive are presented as monthly averages for each testing strategy. Yields are presented as proportions (total tested positive/total tested). Descriptive statistics are used to report on linkage and ART initiation within three months of HIV-positive test date, as well as retention in care, defined by an ART visit at least three months after initiation. Time-to-event methods are used to examine ART initiation with Kaplan–Meier curves graphically displaying time from HIV-positive test date to ART initiation. Cox proportional hazards models calculating hazard ratios (HRs) and 95% confidence intervals (CIs) are shown, adjusting for age, sex, and district and stratified by the pre/post Treat All guideline change. All analyses are conducted in Stata/SE 15.1.

## Results

### Pediatric testing, positive tests, and yields by strategy

During M-HIT implementation, an average of 7,351 tests were conducted across all community-based strategies for all age groups, among which 3,399 (46%) tests were on children 0–14 years. An average of 141.65 individuals tested positive each month; 9% were children. Yields were 3.73%, 0.98%, and 0.38% for adults, adolescents (15–19 years), and children, respectively (Table 2). MOCs tested the most children and identified the largest number of HIV-positive children, followed by D2D, Roadside testing, and then D2D Index strategies. The Facility Index and Targeted testing strategies contributed the smallest numbers in terms of average number of children tested and identified as HIV-positive. Semi-Static did not have any HIV-positive results among the children tested. The pediatric yield varied by strategy from 0.00% to 0.66%. During the same time period, MoH facility-based pediatric yields were 0.72% with an average of 8,526 monthly tests conducted in the two districts.

The majority of all children tested in M-HIT were between 5–14 years (55%) and female (54%). Table 3 shows the breakdown for each strategy. All strategies except MOCs were successful in finding older children to test, with 60–94% of the total children tested being over the age of 5. Yields were similar between age groups, at 0.35% and 0.41% for under and over 5 years, respectively. Differences in yield by sex were also negligible, with 0.40% of males testing HIV-positive compared to 0.37% of females.

Previous test history was known for all strategies except MOC. Among those with known HIV testing history, 75% were first-time testers and 86% of children testing HIV-positive had never previously been tested for HIV. When observing test history by age, 81% and 72% of children under and over 5 years were first-time testers, respectively. For those testing positive, 93% of children under 5 had never been tested before, as well as 83% of children over 5.

### Linkage to care

A total of 227 HIV-positive children were identified through all M-HIT strategies in the time period examined, among which 173 (76%) were enrolled in the follow-up cohort. Among

**Table 2. Average monthly number tested, proportion tested, number tested positive, and yields; stratified by age group for all M-HIT strategies.**

| | | Adults | | | | Adolescents | | | | Children | | | |
|---|---|---|---|---|---|---|---|---|---|---|---|---|---|
| | Tested, All Ages* | Tested | Proportion tested | Tested positive | Positive yield | Tested | Proportion tested | Tested positive | Positive yield | Tested | Proportion tested | Tested positive | Positive yield |
| | (N) | (n) | (%) | (n) | (%) | (n) | (%) | (n) | (%) | (n) | (%) | (n) | (%) |
| MOC | 2398 | 1182 | 49% | 30.80 | 2.60% | 137 | 6% | 1.40 | 1.02% | 1079 | 45% | 4.33 | 0.40% |
| D2D | 1708 | 702 | 41% | 36.17 | 5.15% | 207 | 12% | 1.87 | 0.90% | 799 | 47% | 2.27 | 0.28% |
| D2D Index | 313 | 56 | 18% | 4.20 | 7.50% | 28 | 9% | 0.47 | 1.66% | 229 | 73% | 1.50 | 0.66% |
| Roadside | 640 | 322 | 50% | 13.04 | 4.06% | 105 | 16% | 0.92 | 0.88% | 214 | 33% | 0.73 | 0.34% |
| Facility Index | 90 | 17 | 19% | 0.83 | 5.00% | 10 | 11% | 0.17 | 1.65% | 63 | 70% | 0.42 | 0.66% |
| Targeted | 73 | 26 | 35% | 1.63 | 6.34% | 8 | 10% | 0.13 | 1.64% | 40 | 55% | 0.25 | 0.63% |
| Semi-Static | 40 | 23 | 58% | 1.13 | 4.84% | 7 | 18% | 0.00 | 0.00% | 10 | 24% | 0.00 | 0.00% |
| **ALL M-HIT**** | **7351** | **3265** | **44%** | **121.82** | **3.73%** | **687** | **9%** | **6.76** | **0.98%** | **3399** | **46%** | **13.08** | **0.38%** |

*Any differences in sums is due to rounding.

**All M-HIT is the average number from all testing strategies combined, for the average number of months each strategy was implemented for.

those enrolled, 20% were 0–1 years; 24% 2–4 years; 35% 5–9 years; and 21% 10–14 years. Seventy-one percent (123/173) linked to care within three months, with all children except one initiating ART. Eighty-two percent of those initiating were still engaged in care at three months. An additional 14 children linked to care within six months and five children linked to care before the study concluded, for an overall linkage outcome of 82%. All 19 of these children also initiated ART.

When the ART guidelines changed nine months into the project to adopt the Treat All policy, there was an expected increase in ART initiations as all children became eligible (eligibility was not documented prior to policy change and it cannot be determined whether eligible initiation rates did increase). Fifty-seven percent of children initiated before Treat All compared to 70% after (p = 0.002). However, children engaged in care at three months was comparable before and after Treat All (85% vs 81%, p = 0.602). The median (Interquartile Range) time to initiation was 25 (6–51) days before Treat All and 2 (0–19) days after (p<0.001).

By design all enrolled children should have received four e-vouchers; however, only 74% did due to technical challenges. This allowed for a natural experiment to assess whether receipt of e-vouchers affected linkage to care or ART initiation. As shown in Fig 1, prior to Treat All implementation there was no statistically significant difference in the time to ART initiation between those that did and did not receive a transportation voucher (p = 0.429). However, after Treat All those that received a voucher were more likely to both initiate ART and initiate more quickly (p = 0.007). The adjusted HR for children after Treat All was implemented was statistically significant for those receiving a voucher (aHR: 2.00, 95% CI 1.18–3.38). This means that children who received a voucher had a rate of ART initiation two times faster than children who did not receive the voucher.

## Discussion

M-HIT strategies were successful at finding high proportions of children to test within targeted and blanket strategies. All strategies had lower yields than expected–under 1%–but facility yields were also under 1% during the same period. It is a commonly reported outcome among

**Table 3. Average monthly number tested, proportion tested, number tested positive, and yields among children; stratified by age and sex.**

| | Tested, All Children* | 0–4 yrs | | | | 5–14 yrs | | | | Males | | | | Females | | | |
|---|---|---|---|---|---|---|---|---|---|---|---|---|---|---|---|---|---|
| | | Tested | Proportion tested | Tested positive | Positive yield | Tested | Proportion tested | Tested positive | Positive yield | Tested | Proportion tested | Tested positive | Positive yield | Tested | Proportion tested | Tested positive | Positive yield |
| | (N) | (n) | (%) | (n) | (%) | (n) | (%) | (n) | (%) | (n) | (%) | (n) | (%) | (n) | (%) | (n) | (%) |
| MOC | 1079 | 680 | 63% | 2.37 | 0.35% | 399 | 37% | 1.97 | 0.49% | 488 | 45% | 2.17 | 0.44% | 590 | 55% | 2.17 | 0.37% |
| D2D | 799 | 235 | 29% | 0.50 | 0.21% | 565 | 71% | 1.77 | 0.31% | 370 | 46% | 1.17 | 0.32% | 430 | 54% | 1.10 | 0.26% |
| D2D Index | 229 | 76 | 33% | 0.63 | 0.83% | 152 | 67% | 0.87 | 0.57% | 109 | 48% | 0.63 | 0.58% | 120 | 52% | 0.87 | 0.72% |
| Roadside | 214 | 54 | 25% | 0.07 | 0.12% | 160 | 75% | 0.57 | 0.35% | 94 | 44% | 0.23 | 0.25% | 120 | 56% | 0.40 | 0.33% |
| Facility Index | 63 | 20 | 32% | 0.10 | 0.49% | 43 | 68% | 0.07 | 0.16% | 29 | 46% | 0.07 | 0.23% | 34 | 54% | 0.10 | 0.29% |
| Targeted | 40 | 16 | 40% | 0.03 | 0.21% | 24 | 60% | 0.03 | 0.14% | 17 | 42% | 0.00 | 0.00% | 23 | 58% | 0.07 | 0.29% |
| Semi-Static | 10 | 1 | 6% | 0.00 | 0.00% | 9 | 94% | 0.00 | 0.00% | 5 | 51% | 0.00 | 0.00% | 5 | 49% | 0.00 | 0.00% |
| **ALL M-HIT** | **3399** | **1532** | **45%** | **5.40** | **0.35%** | **1867** | **55%** | **7.68** | **0.41%** | **1555** | **46%** | **6.22** | **0.40%** | **1844** | **54%** | **6.85** | **0.37%** |
| **Facility-based** | 8526 | 3475 | 41% | 24.97 | 0.72% | 5050 | 59% | 36.70 | 0.73% | data not available | | | | | | | |

* Any differences in sums is due to rounding.

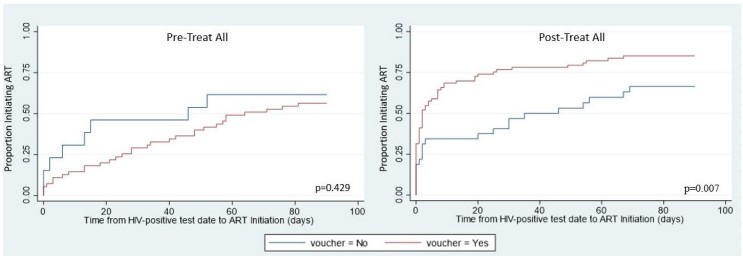

**Fig 1. Kaplan-Meier curves for time to ART initiation by voucher status: Pre and post-Treat All.**

research in this region to find facility yields higher than those in the community [18, 28]. Another CBT focused study conducted in Lesotho in an overlapping time period had a community index testing yield of 1.4% for children and among their other CBT strategies a yield of 0.4% [34].

These low yields were less surprising after the results of Lesotho's Population-Based HIV Impact Assessment (LePHIA) were released in the beginning of 2019 [29]. The survey was conducted between November 2016 to May 2017 and found a 2.1% HIV prevalence among children 0–14 years. The study also reported that Lesotho's pediatric 90-90-90 goals were much closer to being achieved than previous estimates indicated–as reported by parents or guardians 81.8% of children knew their status, 98.2% who knew their status were on ART, and 73.9% of those on ART were virally suppressed.

The M-HIT strategies were effective at reaching previously untested HIV-positive children. This is critical in closing the gap in Sub-Saharan Africa as many studies reveal that a large portion of HIV-exposed children slip through the PMTCT cascade and are subsequently never tested by two years of age [8, 15, 17, 19, 35, 36]. One study in South Africa found that 38% of HTS eligible infants were never tested at immunization clinics that provided EID services [8]. Separate research conducted in a high prevalence setting in South Africa found a high proportion (59%) of untested preschool-aged children despite many with HIV-positive mothers already receiving ART [19]. These mothers reported that after missing ANC testing opportunities they never encountered another systematic testing touchpoint for their children. These studies help explain the existence of a large cohort of undiagnosed older children that often cannot be reached through facility-based HTS. The high proportion of HIV-positive children found in M-HIT over the age of five who had never been tested (83%) supports this assertation and provides evidence that community-based HTS is a promising, yet labor intensive, method for reaching this elusive cohort.

MOCs, the only strategy to offer integrated health services, tested a monthly average of 26% more children than the second highest strategy, D2D, and nearly 80% more than all other strategies. This suggests the widespread success observed in expanding facility-based HTS among young children through offering multiple services at the same delivery point (e.g. immunization) can and should be transferred to a community setting [37]. The majority of children tested at MOCs were under 5 years, which is unsurprising given the package of services (ANC and immunization) that was designed to attract this age group. The high testing numbers of MOCs is noteworthy considering it was the only M-HIT strategy that had to be sought out by caregivers as opposed to the other strategies where caregivers passively accepted testing as it was brought to their homes, daycare centers, or public spaces.

From a holistic health system perspective, MOCs had the greatest overall impact for the clients and the associated health facilities and is recommended for implementation when resources allow. Clients experienced time and cost savings by not having to travel to distant

health facilities to receive services. Anecdotally, healthcare workers also conveyed that MOCs benefited the overall health system by contributing to the decongestion of health facilities, allowing them to provide higher quality and more efficient services at both service points.

Home-based strategies, such as D2D and D2D Index are optimal in settings that strive to achieve comprehensive HIV testing coverage. It effectively brought HTS to individuals of all ages, especially children, which is consistent with other studies [17, 18, 23]. They were successful at finding HIV-positive individuals of all ages and importantly, required limited planning, infrastructure, support, and low level cadres of healthcare workers. Other studies that investigated the effects of home-based HTS found that in addition to consistently high testing uptake rates [7, 21, 22, 38–40], it was successful at overcoming barriers related to stigma [38], and increasing coverage of HTS among the poorest households that are disproportionately not accessing health services at facilities [41].

Similar to other studies, M-HIT's index testing strategies were successful at reaching high proportions of children with relatively higher yields [7, 34, 42]. Facility-based index testing proved to be more resource intensive and inefficient than D2D Index as it required a dedicated room at a facility, the clustering of households once consent was received from ART patients, and the commonly reported occurrence of index leads and household members not being home at the agreed time thus warranting multiple visits. Therefore, it is recommended that future programs in mid to high prevalence settings, especially those targeting the poorest populations, adopt a combination D2D and D2D Index approach. In a low prevalence setting that does not seek high HTS coverage, the targeted strategy of Facility Index is likely to produce higher yields and be less resource intensive.

Roadside testing and Semi-Static roadside testing had the lowest proportions of pediatric HTS (33% and 24%, respectively) suggesting that caregivers are reluctant to have their children tested in a public location and are ineffective for programs focused on children. Finally, the Targeted testing strategy only operated in venues with presumed high-risk children; however, it was quickly realized that nearly all these children had already been tested, explaining that strategy's low testing numbers and eventual termination.

One major perceived shortcoming of CBT is that linkage to care and ART initiation outcomes will be low [43]. This concern is likely founded in the fact that ART initiation is already sub-optimal among facility-based testing, as evidenced by a recent systematic review that found linkage rates for adults between 55%-61% [26]. However, when this review exclusively observed CBT outcomes it revealed that with active client follow-up mechanisms, linkage rates are high (95%, 95% CI: 87–98%) compared to those without any active follow-up (26%, 95% CI: 18–36%). M-HIT's high linkage outcomes are congruent with these results suggesting that factors other than HTS setting contribute more heavily to linkage outcomes.

Despite transportation costs being the most commonly cited barrier to linkage for ART patients in a recent systematic review, M-HIT outcomes did not reflect this to be a primary barrier as most clients who did and did not receive vouchers still linked to care and initiated ART [32, 44]. Our findings showed that those who received an e-voucher were more likely to initiate ART and also initiate faster in comparison to those who did not receive an e-voucher in the post-Treat All era. We surmise that the e-voucher made a significant impact only after the policy change because it was easier for caregivers to comprehend the next steps as families knew their children would definitely receive treatment (versus determining eligibility via CD4 test, which may or may result in treatment). However, even the group receiving an e-voucher only achieved 75% initiation by three months.

Most studies examining the impact of financial incentives on HTS coverage, linkage to care, and ART retention utilize conditional incentives. These show promising short-term effects, even with small amounts of money, while long-term effects are still largely unknown

[45, 46]. The majority of this literature focuses on investigating the use of financial incentives within HTS, with many studies consistently and independently linking it with a marked increase in the uptake of services [46–48].

A scarcity of research exists that investigates the impact of financial incentives on linkage to care and retention outcomes [46–49]. Within this research, a major challenge is posed by the limited comparability of studies [46]. This is primarily due to differences in study population and setting–spanning the urban United States [50], drug injection users in India [51], and cohorts in Sub-Saharan Africa [48, 53–56]. Other fundamental differences among these studies include incentive amount and type (cash versus nonmonetary incentives such as mobile phone credit), outcome objectives (linkage to care, ART initiation, and 12-month retention in care), as well as the conditionality and timing of incentive distribution to clients. Another primary challenge is that most of this research investigates combination interventions that do not allow the effect of financial incentives to be measured in isolation [46, 47, 52–54]. Inconsistent results were reported across studies providing limited generalizable evidence for implementors and policy makers. One of their few commonalities, also echoed in this paper, is the expressed need for further research that uncovers optimal models for financial incentives within linkage to care and retention interventions [46–55].

Despite literature prior to Treat All showing ART initiation rates are often much lower than linkage rates [45], we discovered that once clients linked to care, almost all initiated on ART. After the implementation of Treat All, the median time from testing to ART initiation reduced by 80% to 2 days, indicating people sought care almost immediately. The adoption of Treat All shows great promise for achieving the second and third 90-90-90 targets when considering these outcomes alongside other studies with consistent results [56] and those finding improvements in 12-month retention in care [57]. However, it is also widely acknowledged that these policy changes must be accompanied by health system strengthening to realize their optimal impact [56–59].

## Challenges

There were several limitations to this evaluation. First, we were assessing strategies that were part of ongoing implementation work through the MoH and their partners. As such, this could be considered implementation research whose advantage of the real-world setting effectively identifies feasibility challenges and seeks to improve programs [60]. However, the disadvantage is working outside of a controlled study setting. For example, 24% of all children who tested positive through an M-HIT strategy were 'lost', meaning they were not enrolled into the project's mobile application and therefore also excluded from the follow-up cohort. The lack of enrollment occurred due to healthcare worker error and a shortage of mobile phones among staff as the project rapidly scaled-up during certain time periods. Additionally, some children did not receive an e-voucher due to network failures in remote locations, client errors in M-Pesa registration, or general system errors. Distance to clinic may have been a confounding factor with linkage to care as well, as these same children may have had a more difficult time getting to the nearest health facility for services.

MOCs operated through government funded health facilities who also supplied most commodities and medicines. Challenges were faced in the consistent provision of these materials. Additionally, the local communities of each MOC were responsible for providing a venue to host the clinics. This often resulted in inadequate spaces that lacked privacy for clients to receive sensitive services. Many also lacked an indoor waiting area that proved problematic in the winter months, with some MOCs being canceled due to inclement weather.

Another challenge was that wide variations in national pediatric HIV estimates were experienced throughout the life of the evaluation (19,000 in 2015; 13,000 in 2016; and 16,000 in 2017). This affected project targets and the general understanding of the pediatric HIV landscape in Lesotho [31].

One limitation on the analysis was the spatial and time overlap among the community and facility-based testing, which made it impossible to compare between strategies; nor could it be concluded whether a child tested in the community would have eventually been tested at the facility. Given that the testing strategies were being implemented in overlapping areas, disentangling the effects of each one alone and from facility-based testing would require more variation in the intensity of the implementation of the strategies over space and time. The small number of children testing HIV-positive also limited our power to detect differences.

An important consideration in determining the feasibility of CBT is to understand that resources and costs associated with such work. This paper does not address these areas in any detail; however, it is highly recommended these be thoroughly explored for each potential strategy as costs can vary greatly.

## Conclusion

In Lesotho, identifying HIV-positive children is rare, making each positive test important. Facility-based testing must remain the backbone of HTS, however community-based strategies have proven crucial in closing the gap and finding HIV-positive children. This is especially true in finding those who had never been tested before and with children over two years of age who have limited touchpoints with the formal healthcare system. Linkage to care and ART initiation outcomes were high, which is largely attributed to the active follow-up of clients. Retention in care at three months dropped by almost 20%, indicating continual support is needed. Overall, these CBT strategies contributed substantially to the local HTS landscape suggesting that with adequate resources, community-based HTS is scalable and can have a substantial impact on the health system. While this study adopted Lesotho's Know Your Status policy–offering HTS to anyone with an unknown status–a critical next step toward achieving universal ART coverage is the investigation of more efficient CBT approaches.

## Supporting information

**S1 Dataset.**
(XLSX)

**S2 Dataset.**
(XLSX)

**S1 File.**
(PDF)

## Acknowledgments

We would like to thank the implementing partners—Lesotho's Ministry of Health, Baylor College of Medicine Children's Foundation–Lesotho, Population Services International Lesotho, and Riders for Health Lesotho for their tireless efforts in supporting the provision of high-quality medical services. Finally, we would like to thank the study participants.

## Author Contributions

**Conceptualization:** Kathleen Sindelar, Fumane Lekoala, Matsitso Mohoanyane, Jill Sanders, Jessica Joseph.

**Data curation:** Fumane Lekoala.

**Formal analysis:** Chivimbiso Maponga, Jessica Joseph.

**Investigation:** Kathleen Sindelar, Jill Sanders, Jessica Joseph.

**Methodology:** Kathleen Sindelar, Fumane Lekoala, Jill Sanders, Jessica Joseph.

**Project administration:** Kathleen Sindelar, Chivimbiso Maponga, Esther Mandara, Matsitso Mohoanyane, Jill Sanders.

**Resources:** Esther Mandara.

**Supervision:** Kathleen Sindelar, Chivimbiso Maponga, Fumane Lekoala, Esther Mandara, Matsitso Mohoanyane, Jill Sanders, Jessica Joseph.

**Validation:** Kathleen Sindelar, Jill Sanders, Jessica Joseph.

**Visualization:** Kathleen Sindelar, Jessica Joseph.

**Writing – original draft:** Kathleen Sindelar.

**Writing – review & editing:** Kathleen Sindelar, Chivimbiso Maponga, Fumane Lekoala, Esther Mandara, Matsitso Mohoanyane, Jill Sanders, Jessica Joseph.

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
