## [Decision Letter · Decision Letter 0]

22 Oct 2019

PONE-D-19-26812

Beyond the Facility: An Evaluation of Seven Community-Based Pediatric HIV Testing Strategies and Linkage to Care Outcomes in a High Prevalence, Resource-Limited Setting

PLOS ONE

Dear Ms Sindelar,

Thank you for submitting your manuscript to PLOS ONE. After careful consideration, we feel that it has merit but does not fully meet PLOS ONE’s publication criteria as it currently stands. Therefore, we invite you to submit a revised version of the manuscript that addresses the points raised during the review process.

We would appreciate receiving your revised manuscript by Dec 06 2019 11:59PM. To enhance the reproducibility of your results, we recommend that if applicable you deposit your laboratory protocols in protocols.io, where a protocol can be assigned its own identifier (DOI) such that it can be cited independently in the future. For instructions see: http://journals.plos.org/plosone/s/submission-guidelines#loc-laboratory-protocols

We look forward to receiving your revised manuscript.

Kind regards,

Marcel Yotebieng, M.D., MPH, Ph.D

Academic Editor

PLOS ONE

Journal Requirements:

2. In ethics statement in the manuscript and in the online submission form, please provide additional information about the patient records used in your retrospective study. Specifically, please ensure that you have discussed whether all data were fully anonymized before you accessed them and/or whether the IRB or ethics committee waived the requirement for informed consent. If patients provided informed written consent to have data from their medical records used in research, please include this information.

4. Thank you for stating the following in the Financial Disclosure section:

'This project was made possible through a consortium of funders that was headed by the Vodafone Foundation. Other funders include the U.S. Agency for International

Development (USAID), the ELMA Foundation, Elton John AIDS Foundation, and Viiv

Healthcare. No grant numbers were assigned for this project. None of the

aforementioned funders played a role in the design of the study, metrics measured,

data collection and analysis, decision to publish, or the preparation of the manuscript.'

We note that you received funding from a commercial source: Vodafone Foundation.

Additional Editor Comments (if provided):

The reviewers all agreed that this is a promising data-set. however they are also all in agreement that the manuscript as written currently is not technically sound. Please as you prepare a revision, use the STROBE Checklist to ensure that all important require element are included at the appropriate place.

Reviewers' comments:

Reviewer's Responses to Questions

**Comments to the Author**

1. Is the manuscript technically sound, and do the data support the conclusions?

Reviewer #1: No

Reviewer #2: No

Reviewer #3: Yes

2. Has the statistical analysis been performed appropriately and rigorously? 

Reviewer #1: N/A

Reviewer #2: No

Reviewer #3: Yes

3. Have the authors made all data underlying the findings in their manuscript fully available?

Reviewer #1: Yes

Reviewer #2: Yes

Reviewer #3: Yes

4. Is the manuscript presented in an intelligible fashion and written in standard English?

Reviewer #1: Yes

Reviewer #2: Yes

Reviewer #3: Yes

5. Review Comments to the Author

Reviewer #1: There are three main issues I feel need to be addressed before I can commit to a full review of this paper:

1. Although the authors note that M-HIT was a collaborative effort among five partners, including the Lesotho MOH, all of the authors are from CHAI. Including the IPs and MOH in production may have assisted with the two more serious issues that need to be addressed below.

2. A description clearly describing the assent/consent process for the children included in this testing initiative is needed.

3. A clear description of the eligibility criteria for testing is needed to assess the results of the paper. Were children of HIV-negative mothers tested in this program? Were children tested without first assessing the status of the mother? There are situations in which that may be appropriate. However, it needs to be clearly described as it has a big impact on how we can assess the results. Please also comment on whether the eligibility criteria for testing is in line with Lesotho MOH policy or whether there was a deviation from standard practice for the purpose of this study.

Reviewer #2: Summary

This article by Sindelar et al, presents a large, unique, and important dataset from Lesotho and compares several different strategies for pediatric HIV testing in community-based settings. The dataset, drawn from a large collaborative implementation program, is valuable and compares head-to-head several strategies for pediatric testing which are often presented separately. As pediatric HIV testing continues to scale up in facilities for older children outside of PMTCT settings, critically examining the comparative effectiveness of different testing strategies is essential. In addition to considering testing coverage/uptake and yield, which are often the focus of case detection strategies, this paper also considers linkage to care and ART initiation, which makes it a valuable addition. However, there are numerous large methodologic issues in data interpretation, presentation, and contextualization that make the current version of the manuscript not scientifically sound. Revising the analysis and interpretation could transform this promising dataset into important lessons for those in the field of pediatric HIV testing. I enjoyed reading this paper and would look forward to reviewing a revision.

Major essential comments

1. Concepts of uptake of testing (proportion testing among those approached), testing yield (proportion testing positive among those testing), and relative numbers testing and testing positive between different strategies and age bands are reported inconsistently throughout manuscript (methods, results, discussion) but could be revised for clarity. Below are specific notes of where these concepts seem to be misinterpreted or inconsistently applied. Suggest reporting % uptake and % yield for each strategy, along with the already presented numbers testing and testing positive for each strategy. Further suggest adjusting these metrics to be per month to address differences in time period of data collection. Current reporting and comparisons are not scientifically valid.

a. Lines 134-138: suggest aligning these definitions of outcomes with the aforementioned % uptake, % yield, number tested, and number tested positive. Suggest providing numerator and denominator data (as appropriate) for each outcome. “Contribution of community-based testing…” is unclear as a metric. Does this mean the proportion of all children tested within the program that came from each strategy?

b. Line 160: unclear what testing volumes and identifications mean, but likely correspond to number tested and number testing positive; the terminology here differs from that mentioned earlier in the methods in lines 134-138. Suggest harmonizing even if my suggested terminology above is not your preferred terminology.

c. Lines 173-177: “MOCs tested and identified the highest number of children…contributed the smallest numbers in terms of overall children tested and identified, partly due to their shorter implementation timeframe”. Suggest adjusting all of the estimates to be per month of time, otherwise it is not a fair or valid comparison.

d. Lines 212-215: New data are introduced here (e.g. “index case testing which conducted over 80% of all tests on children”) and reflect new metrics that are not described in the results. Is this maybe referring to uptake of testing among those tested? The proportions given in this section do not add to 100%, so don’t seem to be relative contribution to numbers tested or yield. Also see lines 234-235 for 33% and 24% numbers which seem to be related.

2. In addition to defining and utilizing terms consistently as mentioned above, suggest harmonizing language about comparisons that are made and what they mean, and using precise language (e.g. avoid “significant” unless in the statistical sense, avoid terms like “incredibly effective” [line 216] if there is no comparison group or reference).

a. Line 214: Suggest not using significant unless these were statistically compared; later in discussion in lines 285-286 it says it was “not possible to compare between strategies” including community and facility-based.

b. Line 179-180: In several places, the article describes the proportion of children who were ages 5-14 among those who tested positive. While this is not an incorrect proportion to calculate, it cannot be interpreted as it is in the discussion to reflect prevalence or yield being higher in different groups. Suggest including the age breakdown among those children tested, not just those who test positive to identify whether yield differs between the groups. If n/N and % are included for each metric, all of the important absolute and relative comparisons can be made.

c. Lines 225-226: It’s not accurate to say that the greater number of children identified as positive using MOCs suggests that “children are more likely to receive HTC when it is provided alongside other health services”. That would be supported by a comparison of uptake between settings, not the relative number of positives between strategies.

d. Lines 239-242: It is not possible to evaluate the accuracy of the first sentence here about the contribution of CBT being relatively more than facility-based, as the authors do not present overall numbers in the results. This statement also contradicts other sections of the manuscript which say that facility based testing still has more positive children identified and higher uptake.

3. Linkage to care and ART initiation data are wonderful to see but should have a time period and window period in their definitions (e.g. linkage to care within 90 days after HIV diagnosis, or presented for scheduled visit within 30 days). Suggest revising lines 139-146 to include both, and also define how “active” was defined in determining who was retained.

4. Contextualizing these testing strategies more accurately and precisely within the peds testing literature in terms of uptake, yield, and linkage would help influence decision-making. For example, these data support what we’ve seen in other literature that index case and targeted testing strategies tend to have higher yield than “blanket” testing models, but the discussion in lines 225-252 lacks clarity in interpretation. There are important lessons on how to choose between these strategies, but they are not clearly presented.

5. Lines 203-204: This is a very important analysis that tests an important question of whether financial assistance/incentives/support modifies linkage to care. Given the sample size of positives in this study, it would be crucial to see the data or the point estimates and 95%CI for this analysis in order to conclude that there were no differences. Additionally, later in the paper we see that e-vouchers didn’t make it to families for a variety of reasons, some of which are reasonable to expect are random (e.g. mpesa registration issues) and others which would be expected to be related to the outcome of linkage (e.g. distance to clinic). Therefore, there is a clear potential for confounding (lines 266-267), which should be addressed and presented in this important analysis.

Minor important comments

1. Lines 107-108: suggest revision to “No sampling procedure was implemented as this program evaluation was conducted to evaluate routine implementation”. Implementation research or implementation science is a scientific discipline that often employs sampling procedures to test hypotheses, so would not be an accurate rationale for not including a sampling process.

2. Table 1: Unclear what the factors were that were used to determine “high risk of HIV infection”; would be crucial to define what those are and how they were identified and operationalized. With risk screening, a major barrier is often coverage of risk assessment, so would be good to comment on this.

3. Lines 162-164: Comparing time to linkage to care and ART initiation should be done using survival analysis (Cox proportional hazards regression, or some other version), not Wilcoxon’s rank sum (which is appropriate for comparing continuous time where all individuals have completed the outcome). Suggest revising the statistical test used to compare these. Did the authors perhaps intend to use a log rank test instead of a rank sum for survival data that do not have proportional hazards?

4. Line 164-165: For the negative binomial regression, suggest including a bit more information on the model itself and how it accounted for the varying length of time at each setting, and how it incorporated facility-based testing. Unclear as is. Additionally, suggest rechecking model results in lines 187-193, which do not reflect what would be expected based on the yield of each testing strategy.

5. Line 249: related to above, important to include in this sentence that the linkage to care intervention included provision of financial assistance, which is not standard in many settings.

6. New data are introduced on lines 227-230. This is really important and valuable information that sheds light on implementation challenges. Should be included in methods and results as data, not in discussion.

7. Glad to see the acknowledgement of need to have cost data in lines 294-295!!! This is really notably missing from the pediatric HIV testing literature and is a crucial decision-making factor for MOH. Suggest that authors consider writing a follow-up paper that documents any cost data they have, even if only number of staff members required to be employed to reach these numbers.

Discretionary comments

1. Abstract: suggest clarifying time period for linkage and ART initiation and suggest providing N and n for the yields.

2. Figure 1 and text do not match one another in terms of concepts being explained and relevant denominators.

3. Line 198: what were the treatment guidelines for children during this period of time and could the time to ART initiation be confounded by the changes in guidelines that are mentioned in the discussion?

4. Lines 204-206: Interesting finding about ART initiation and e-vouchers; suggest explaining why this might exist in discussion in lines 258-262.

5. Lines 219-220: While I agree that community-based testing likely does identify children earlier in their disease progression than facility-based testing, the present manuscript doesn’t have data to support this; suggest rephrasing this sentence to point more to other literature instead of stating as a lesson from these data.

6. Lines 221-224: there are many studies in this space, several systematic reviews and broader synthesis papers, and global datasets and national surveys that support this point. Suggest including additional information to contextualize more broadly.

7. Line 250: who is the reference group for this?

8. Line 256: There are systematic reviews about the effect of financial assistance on linkage; suggest offering a broader review of this literature to more appropriately contextualize.

9. Line 263: not possible to evaluate how recent the data in this reference are as ref is missing; were the data before the era of test and treat all? That could explain gaps. If mixed, suggest restricting to time period of post test and treat so that there shouldn’t be a gap.

10. Line 274: why were children “lost”? this is a large proportion and could influence estimates of linkage and treatment and retention.

11. Line 280: ANC services? HIV services?

Reviewer #3: Abstract:

• Instead of saying “Majority of children identified were…” include the exact proportion of children identified

• It may be better to put the number of children diagnosed rather than 9% of the overall number of people as the paper is about paediatric testing

• Abstract doesn’t mention of the 7 strategies, however these where in the title of the paper?

• Was ART initiated and dispensed in the community for the duration of the follow up period? May need to clarify in the abstract where ART was dispensed. Title and abstract implies that ART was dispensed in community settings.

• Line 33 are “enrolled” children the same as those children diagnosed?

Introduction

1. Line 81: What year is this data from?

2. The introduction would benefit form mention of subsequent linkage to care concerns of community-based HIV testing strategies

3. The introduction outlies in great detail the barriers to testing and HIV testing gaps among children but does not bring forth any evidence of how community-based HIV testing could bridge this gap? Has previous work on community-based testing for children been done in other settings? Has this been effective?

4. The introduction provides a lot of data on ART coverage, however, is there is any data of HIV testing coverage in the age group? What is covered of early infant diagnosis in Lesotho or the region? Or PMTCT coverage? This data will be useful to give context for the testing gaps that exist if available.

5. Line 81: the prevalence is 2.1% among children of what age range?

6. Line 90: is there a reference to M-HIT anywhere? A protocol? This would be helpful for additional context to the study

Methods

7. Is this paper presenting M-HIT or a subcomponent of M-HIT? It may be important to clarify that this paper is focusing on apart of M-HIT and clearly demarcate which part. This is not clear.

8. Line 104 that says the testing strategies were focused on rural areas is contradictory to the rest of the section. Are Maseru and Leribe rural areas?

9. Table 1: The different testing strategies are unclear in their current format. It may be useful if there was some expansion to these as the title of the paper is an evaluation of the 7 strategies and they therefore should play a major role in the paper. E.g. *Targeted testing: what was the selection criteria for a child who is high risk? Where are the venues described? Schools? Shopping complexes? Community halls?

*The relationship between facility index testing, door to door testing and door to door index testing is not clear. If testing is offered for all household members in facility index testing, how then do you index an individual in the same household become an index when everyone in that household has been tested already?

10. Line124: This appears as another intervention in addition to just community-based testing leading to ART initiation i.e. another motivator (economic) was provided which may have resulted in the subsequent high linkage and retention rates.

11. Line 137: Please consider rewording to “Diagnosing children unknown HIV”. Identifying implies that the testing may have been done prior to you finding the child.

12. 138: You would need a denominator to quantify this i.e. your denominator would be total number of children tested in each district (community and facility), however, this is not presented in the results?

13. Line 145: Describing the expected ART visits in the 3 months may be useful in order for the reader to understand how many times a client would have come within the 3 months e.g. are they just supposed to come to the clinic once in the 3 months to be classified as retained?

14. What is the justification of only 3-month retention given that ART is a lifelong treatment?

Results

1. Line 172: It would make more sense to present the yield of HIV for each testing strategy as each testing strategy was not implemented for the same amount of time. That would make a better argument for cost effectiveness of each strategy. As a stand-alone this figure is not presenting that much additional data aside from the breakdown in diagnosis by age in the different testing strategies. It may be better presented in a table that presents additional demographic information such as Gender of the children.

2. Is additional demographic information such as gender available? This would useful to add to the paper

3. Line 196: What happened to the other children that were not enrolled? What was the reason for non-enrolment? Do we know if they were subsequently linked to care as including the non-enrolled in the denominator would affect the linkage to care proportion significantly?

4. Line 204: why are there no results shown? As noted above this result is critical in understanding weather or not the e-voucher impacted linkage to care.

Discussion

1. Line 248: the timely client follow-up support is not previously defined? How was this different from facility care?

2. Line 247: it will be best if the proportion of children lost be included in the results as mentioned above. How was “lost” defined? Is this that there was no record of them at the clinic when you followed up? If so that may be better categorised as “not linked to care”? Again, the follow up procedure between diagnosis to linkage should be clearly described in the paper.

6. PLOS authors have the option to publish the peer review history of their article (what does this mean?). If published, this will include your full peer review and any attached files.

Reviewer #1: No

Reviewer #2: No

Reviewer #3: Yes: Chido Dziva Chikwari

---

## [Author Response · Author response to Decision Letter 0]

17 Dec 2019

Reviewer #1: 

There are three main issues I feel need to be addressed before I can commit to a full review of this paper:

1. Although the authors note that M-HIT was a collaborative effort among five partners, including the Lesotho MOH, all of the authors are from CHAI. Including the IPs and MOH in production may have assisted with the two more serious issues that need to be addressed below.

We agree with this statement, and as a result reached out to the implementing partners (including the Ministry of Health) to request their participation in authoring this paper. Two new co-authors were added, each of whom were involved in the design and supervision of the project and have assisted in clarifying your highlighted concerns.

2. A description clearly describing the assent/consent process for the children included in this testing initiative is needed.

This has been added in the paper under the Testing and Follow-Up Procedures subsection and have highlighted its alignment with MoH guidelines. Further clarification was provided on lines 209 - 214 to explain that both the local and international IRBs did not require written consent for the evaluation.

3. A clear description of the eligibility criteria for testing is needed to assess the results of the paper. Were children of HIV-negative mothers tested in this program? Were children tested without first assessing the status of the mother? There are situations in which that may be appropriate. However, it needs to be clearly described as it has a big impact on how we can assess the results. Please also comment on whether the eligibility criteria for testing is in line with Lesotho MOH policy or whether there was a deviation from standard practice for the purpose of this study.

All points have been outlined in the Testing and Follow-Up Procedures subsection (lines 155 – 182).

Reviewer #2: 

Summary

This article by Sindelar et al, presents a large, unique, and important dataset from Lesotho and compares several different strategies for pediatric HIV testing in community-based settings. The dataset, drawn from a large collaborative implementation program, is valuable and compares head-to-head several strategies for pediatric testing which are often presented separately. As pediatric HIV testing continues to scale up in facilities for older children outside of PMTCT settings, critically examining the comparative effectiveness of different testing strategies is essential. In addition to considering testing coverage/uptake and yield, which are often the focus of case detection strategies, this paper also considers linkage to care and ART initiation, which makes it a valuable addition. However, there are numerous large methodologic issues in data interpretation, presentation, and contextualization that make the current version of the manuscript not scientifically sound. Revising the analysis and interpretation could transform this promising dataset into important lessons for those in the field of pediatric HIV testing. I enjoyed reading this paper and would look forward to reviewing a revision.

Major essential comments

1. Concepts of uptake of testing (proportion testing among those approached), testing yield (proportion testing positive among those testing), and relative numbers testing and testing positive between different strategies and age bands are reported inconsistently throughout manuscript (methods, results, discussion) but could be revised for clarity. Below are specific notes of where these concepts seem to be misinterpreted or inconsistently applied. Suggest reporting % uptake and % yield for each strategy, along with the already presented numbers testing and testing positive for each strategy. Further suggest adjusting these metrics to be per month to address differences in time period of data collection. Current reporting and comparisons are not scientifically valid.

We agree that uptake is an important metric, however this was not included in the study design and consequently was only measured for one of the strategies. Since we were unable to calculate uptake for all strategies, it was not included.

We have adopted your suggestion to report all testing volumes by monthly average and adjusted all results for each strategy. Objective 1 (lines 114-116) provides a description for the term “yield” and the presentation of all testing numbers have been revised.

a. Lines 134-138: suggest aligning these definitions of outcomes with the aforementioned % uptake, % yield, number tested, and number tested positive. Suggest providing numerator and denominator data (as appropriate) for each outcome. “Contribution of community-based testing…” is unclear as a metric. Does this mean the proportion of all children tested within the program that came from each strategy?

We have revised the objectives accordingly and aligned terms as suggested. In addition, we have revised three of the visuals, replacing the original Figure 1 with Table 2, the original Figure 2 with Table 3, and the original Figure 4 with Figure 2. We have included facility-based testing numbers in the greatest detail that they were available in DHIS2 (the MoH data warehouse). Tables now include a clearer description of the indicators being presented especially percentages/proportions. 

 “Contribution of community-based testing…” was eliminated to avoid confusion – after reconsideration the authors felt it was not a necessary or helpful metric to highlight. 

b. Line 160: unclear what testing volumes and identifications mean, but likely correspond to number tested and number testing positive; the terminology here differs from that mentioned earlier in the methods in lines 134-138. Suggest harmonizing even if my suggested terminology above is not your preferred terminology.

Most uses of the word “identification” have been adjusted to “positive test”. Those which were not changed were given increased context for clarity. All use of the word “testing volumes” have been replaced to “testing numbers”.

c. Lines 173-177: “MOCs tested and identified the highest number of children…contributed the smallest numbers in terms of overall children tested and identified, partly due to their shorter implementation timeframe”. Suggest adjusting all of the estimates to be per month of time, otherwise it is not a fair or valid comparison.

Agreed, as previously mentioned all numbers presented for each testing strategy are now a monthly average (described in lines 216-217).

d. Lines 212-215: New data are introduced here (e.g. “index case testing which conducted over 80% of all tests on children”) and reflect new metrics that are not described in the results. Is this maybe referring to uptake of testing among those tested? The proportions given in this section do not add to 100%, so don’t seem to be relative contribution to numbers tested or yield. Also see lines 234-235 for 33% and 24% numbers which seem to be related.

Tables 2 and 3 now display the total number tested, the proportion of tests conducted in each age category, and the yields. This aligns with the written results and clarifies the proportions being presented.

2. In addition to defining and utilizing terms consistently as mentioned above, suggest harmonizing language about comparisons that are made and what they mean, and using precise language (e.g. avoid “significant” unless in the statistical sense, avoid terms like “incredibly effective” [line 216] if there is no comparison group or reference).

a. Line 214: Suggest not using significant unless these were statistically compared; later in discussion in lines 285-286 it says it was “not possible to compare between strategies” including community and facility-based.

All uses of the word significant were checked to ensure this referred to statistical significance. Other superfluous verbiage (i.e. “incredibly effective”) has also been removed or replaced.

b. Line 179-180: In several places, the article describes the proportion of children who were ages 5-14 among those who tested positive. While this is not an incorrect proportion to calculate, it cannot be interpreted as it is in the discussion to reflect prevalence or yield being higher in different groups. Suggest including the age breakdown among those children tested, not just those who test positive to identify whether yield differs between the groups. If n/N and % are included for each metric, all of the important absolute and relative comparisons can be made.

This has been included in Table 3.

c. Lines 225-226: It’s not accurate to say that the greater number of children identified as positive using MOCs suggests that “when it is provided alongside other health services”. That would be supported by a comparison of uptake between settings, not the relative number of positives between strategies.

This is a great point. The sentence was revised so the discussion point is backed up by the data and includes other literature for context (lines 313 – 317).

d. Lines 239-242: It is not possible to evaluate the accuracy of the first sentence here about the contribution of CBT being relatively more than facility-based, as the authors do not present overall numbers in the results. This statement also contradicts other sections of the manuscript which say that facility based testing still has more positive children identified and higher uptake.

This point was removed. 

3. Linkage to care and ART initiation data are wonderful to see but should have a time period and window period in their definitions (e.g. linkage to care within 90 days after HIV diagnosis, or presented for scheduled visit within 30 days). Suggest revising lines 139-146 to include both, and also define how “active” was defined in determining who was retained.

A timebound definition for linkage to care and ART initiation was added, as well as a description of what “active” in care means and how it was documented.

4. Contextualizing these testing strategies more accurately and precisely within the peds testing literature in terms of uptake, yield, and linkage would help influence decision-making. For example, these data support what we’ve seen in other literature that index case and targeted testing strategies tend to have higher yield than “blanket” testing models, but the discussion in lines 225-252 lacks clarity in interpretation. There are important lessons on how to choose between these strategies, but they are not clearly presented.

Agreed. The discussion section was revised extensively to include more literature around strategies, challenges, and outcomes within the Sub-Saharan African context.

A column was added in Table 1 that classifies each strategy as targeted or blanket for increased clarity and ease of reference.

5. Lines 203-204: This is a very important analysis that tests an important question of whether financial assistance/incentives/support modifies linkage to care. Given the sample size of positives in this study, it would be crucial to see the data or the point estimates and 95%CI for this analysis in order to conclude that there were no differences. Additionally, later in the paper we see that e-vouchers didn’t make it to families for a variety of reasons, some of which are reasonable to expect are random (e.g. mpesa registration issues) and others which would be expected to be related to the outcome of linkage (e.g. distance to clinic). Therefore, there is a clear potential for confounding (lines 266-267), which should be addressed and presented in this important analysis.

We have revised this analysis in several ways. First, we took your advice and looked at the potential for confounding before and after the Test and Treat/Treat All guidelines were implemented and decided to stratify all analyses by this variable. Second, instead of using median time to ART initiation as the outcome, we switched to a time-to-event analysis and graphed Kaplan-Meier curves, along with Cox Proportional Hazards on receipt of e-voucher or not. In the Cox regression, we looked at various confounders and ended up adjusting for age, sex and district.

Minor important comments

1. Lines 107-108: suggest revision to “No sampling procedure was implemented as this program evaluation was conducted to evaluate routine implementation”. Implementation research or implementation science is a scientific discipline that often employs sampling procedures to test hypotheses, so would not be an accurate rationale for not including a sampling process.

This was rectified by eliminating the sentence that discussed implementation research as part of the rationale for not sampling and explaining that in order to maximize statistical power to detect effects, all available data from the program will be used rather than a sample of program data (lines 133 – 134).

2. Table 1: Unclear what the factors were that were used to determine “high risk of HIV infection”; would be crucial to define what those are and how they were identified and operationalized. With risk screening, a major barrier is often coverage of risk assessment, so would be good to comment on this.

Further explanation and examples were given in Table 1.

3. Lines 162-164: Comparing time to linkage to care and ART initiation should be done using survival analysis (Cox proportional hazards regression, or some other version), not Wilcoxon’s rank sum (which is appropriate for comparing continuous time where all individuals have completed the outcome). Suggest revising the statistical test used to compare these. Did the authors perhaps intend to use a log rank test instead of a rank sum for survival data that do not have proportional hazards?

Based on your comments we agree that a Cox proportional hazards regression would be a more robust test. Assumptions for this regression were checked and met. Methods (lines 226-229) and Results (lines 282-284) were revised accordingly.

4. Line 164-165: For the negative binomial regression, suggest including a bit more information on the model itself and how it accounted for the varying length of time at each setting, and how it incorporated facility-based testing. Unclear as is. Additionally, suggest rechecking model results in lines 187-193, which do not reflect what would be expected based on the yield of each testing strategy.

Thank you for pointing this out. We agree this was unclear and have added additional text in the methods section (lines 219-225). We have also re-checked the model results. As for the yields, our results do match what we'd expect - for instance, for the D2D Index strategy, our predicted number of positives is 4.95 positives per 1000 tests conducted, which is equivalent to a yield of .50%, vs the actual yield for children of 0.66% (see Table 2).

5. Line 249: related to above, important to include in this sentence that the linkage to care intervention included provision of financial assistance, which is not standard in many settings.

This section was completely revised for greater accuracy and outside literature was included for context.

6. New data are introduced on lines 227-230. This is really important and valuable information that sheds light on implementation challenges. Should be included in methods and results as data, not in discussion.

We did not include any of the qualitative survey results in the Results section as the survey was not conducted on a representative sample. We agree it contains valuable information and therefore relevent inputs were included it in the discussion to shed more light on the challenges.

7. Glad to see the acknowledgement of need to have cost data in lines 294-295!!! This is really notably missing from the pediatric HIV testing literature and is a crucial decision-making factor for MOH. Suggest that authors consider writing a follow-up paper that documents any cost data they have, even if only number of staff members required to be employed to reach these numbers.

Thank you, we will strongly consider this recommendation. 

Discretionary comments

1. Abstract: suggest clarifying time period for linkage and ART initiation and suggest providing N and n for the yields.

The time period has been clarified (lines 205 – 208). Due to word limitations only Ns and yield %s have been included in the results section in Table 2; n’s can all be calculated from the information provided.

2. Figure 1 and text do not match one another in terms of concepts being explained and relevant denominators.

Figure 1 has been removed and replaced with Table 3. 

3. Line 198: what were the treatment guidelines for children during this period of time and could the time to ART initiation be confounded by the changes in guidelines that are mentioned in the discussion?

The treatment guidelines and ART eligibility prior to Treat All policy implementation have been included for further clarity (171 – 173).

4. Lines 204-206: Interesting finding about ART initiation and e-vouchers; suggest explaining why this might exist in discussion in lines 258-262.

Analysis was redone to examine the effect of e-vouchers pre/post Treat All (Figure 2). Despite a lot of internal discussion around this no one had a clear understanding of these outcomes occurred (lines 363 – 366). Futher, an extensive literature search was conducted to add more context, but no relevant articles were found.

5. Lines 219-220: While I agree that community-based testing likely does identify children earlier in their disease progression than facility-based testing, the present manuscript doesn’t have data to support this; suggest rephrasing this sentence to point more to other literature instead of stating as a lesson from these data.

Agreed. This has been revised to align better with the data being presented and more literature was added.

6. Lines 221-224: there are many studies in this space, several systematic reviews and broader synthesis papers, and global datasets and national surveys that support this point. Suggest including additional information to contextualize more broadly community testing to reach children sooner and those who slip through PMTCT cascade.

Several specific studies around gaps in PMTCT have been added (lines 300 – 312).

7. Line 250: who is the reference group for this?

This sentence/reference was removed and more appropriate ones were found and discussed in its place.

8. Line 256: There are systematic reviews about the effect of financial assistance on linkage; suggest offering a broader review of this literature to more appropriately contextualize.

Great suggestion – this was done to the best of our ability given the scarcity of research that examines the effect of unconditional cash vouchers (most are conditional). 

9. Line 263: not possible to evaluate how recent the data in this reference are as ref is missing; were the data before the era of test and treat all? That could explain gaps. If mixed, suggest restricting to time period of post test and treat so that there shouldn’t be a gap.

We now present all initiation and retention proportions both before and after Treat All (lines 270-276).

We looked extensively for research that examined the effect of Treat All policy implementation on ART initiation (timing and overall outcomes), however could not find any in the Sub-Saharan African context. This also would’ve been really helpful in providing more context and possible explanations around the increased effect of e-vouchers in ART initiation post-Treat All.

10. Line 274: why were children “lost”? this is a large proportion and could influence estimates of linkage and treatment and retention.

Lost has been defined (lines 395 – 396), as well as its causes in the Challenges subsection in lines 394-397. 

11. Line 280: ANC services? HIV services?

Replaced ANC services with “sensitive” services (line 403).

Reviewer #3: 

Abstract:

• Instead of saying “Majority of children identified were…” include the exact proportion of children identified

We have updated according to your suggestion (line 39).

• It may be better to put the number of children diagnosed rather than 9% of the overall number of people as the paper is about paediatric testing

We have included the average monthly total number of positives and the overall yield for children testing positive.

• Abstract doesn’t mention of the 7 strategies, however these where in the title of the paper?

The abstract states that seven strategies were piloted, however due to word limitations we were not able to highlight all of them individually.

• Was ART initiated and dispensed in the community for the duration of the follow up period? May need to clarify in the abstract where ART was dispensed. Title and abstract implies that ART was dispensed in community settings.

This has been clarified in the abstract (line 35-36) and Testing and Follow-up Procedures subsection.

• Line 33 are “enrolled” children the same as those children diagnosed?

No – “enrolled” children were those that enrolled in the mHealth system after testing HIV-positive through the project. This has been better delineated in the manuscript (lines 34-36) , and the causes for children failing to be enrolled have been addressed in the Challenges section (lines 392-397).

Introduction

1. Line 81: What year is this data from?

The national survey was conducted in 2016 – 2017.

2. The introduction would benefit from mention of subsequent linkage to care concerns of community-based HIV testing strategies

We agree it needs to be addressed, but felt it fit better in our Discussion section so it could be addressed in greater detail and with additional context from other studies.

3. The introduction outlies in great detail the barriers to testing and HIV testing gaps among children but does not bring forth any evidence of how community-based HIV testing could bridge this gap? Has previous work on community-based testing for children been done in other settings? Has this been effective?

Great suggestion, this has been included in lines 300 – 312 and 328 - 337.

4. The introduction provides a lot of data on ART coverage, however, is there is any data of HIV testing coverage in the age group? What is covered of early infant diagnosis in Lesotho or the region? Or PMTCT coverage? This data will be useful to give context for the testing gaps that exist if available.

PMTCT coverage has been included in the introduction in line 98. Additional data from a national survey whose results were only released after the conclusion of the project has been added in the Discussion section to enhance contextual interpretation of the results (lines 294-299).

5. Line 81: the prevalence is 2.1% among children of what age range?

The age range is 0 – 14 years. This has been clarified in the manuscript and moved to the Discussion section where we felt it was more appropriate (line 296).

6. Line 90: is there a reference to M-HIT anywhere? A protocol? This would be helpful for additional context to the study

This is the program’s first publication, and therefore no prior literature exists on it.

Methods

7. Is this paper presenting M-HIT or a subcomponent of M-HIT? It may be important to clarify that this paper is focusing on apart of M-HIT and clearly demarcate which part. This is not clear.

While this paper does not include all the data that was collected for the M-HIT project, it is presenting on the entire project as a whole. The project actually started off with only 4 testing strategies, but as more strategies were added they were included in the evaluation, and thus included here in this paper. The one major component we do not include in this paper is around pregnant and lactating HIV-positive women, which we have now made clear is excluded in lines 148 - 150.

8. Line 104 that says the testing strategies were focused on rural areas is contradictory to the rest of the section. Are Maseru and Leribe rural areas?

The testing strategies initially focused on rural and underserved areas, however throughout the life of the project it expanded throughout the catchment areas of all government funded health facilities eliminating the sole focus on rural and underserved areas. We have removed the rural reference to not confuse the reader.

9. Table 1: The different testing strategies are unclear in their current format. It may be useful if there was some expansion to these as the title of the paper is an evaluation of the 7 strategies and they therefore should play a major role in the paper. E.g. *Targeted testing: what was the selection criteria for a child who is high risk? Where are the venues described? Schools? Shopping complexes? Community halls?

We have revised Table 1: a column was added that classifies each strategy as targeted or blanket for increased clarity and ease of reference and we have re-worded and/or added further context to the descriptions of the strategies to help clarify. 

*The relationship between facility index testing, door to door testing and door to door index testing is not clear. If testing is offered for all household members in facility index testing, how then do you index an individual in the same household become an index when everyone in that household has been tested already?

The descriptions of strategies in Table 1 were updated for increased clarity.

10. Line124: This appears as another intervention in addition to just community-based testing leading to ART initiation i.e. another motivator (economic) was provided which may have resulted in the subsequent high linkage and retention rates.

Yes, you are correct that the mHealth component could have been another motivator. We have consequently revised our analyses to look at receipt of e-voucher and time to ART Initiation (new Figure 2) and addressed with other research in the Discussion section.

11. Line 137: Please consider rewording to “Diagnosing children unknown HIV”. Identifying implies that the testing may have been done prior to you finding the child.

This word selection was purposeful, as after enrollment it was discovered that a small portion of children had been previously diagnosed, however never linked to care or initiated on ART. We have added this text in at lines 163-165.

12. Line 138: You would need a denominator to quantify this i.e. your denominator would be total number of children tested in each district (community and facility), however, this is not presented in the results?

We have removed this objective from the manuscript to avoid confusion and after reconsideration the authors felt it was not a necessary metric to highlight. 

13. Line 145: Describing the expected ART visits in the 3 months may be useful in order for the reader to understand how many times a client would have come within the 3 months e.g. are they just supposed to come to the clinic once in the 3 months to be classified as retained?

We have added in the timeline in lines 176-177. 

14. What is the justification of only 3-month retention given that ART is a lifelong treatment?

While we did follow-up many patients beyond the three month mark, due to restrictions imposed by project timelines three month retention was the only unbiased time period for retention we could look at (as those that were enrolled at the end of the study did not allow longer than this time period of follow-up to examine, it would not present a true picture of 6- or 12-month retention). Although initially we did report on 6-month retention (among those who had 6-months of follow-up time in) a smaller cohort, we felt this added confusion and therefore removed it. 

1. Line 172: It would make more sense to present the yield of HIV for each testing strategy as each testing strategy was not implemented for the same amount of time. That would make a better argument for cost effectiveness of each strategy. As a stand-alone this figure is not presenting that much additional data aside from the breakdown in diagnosis by age in the different testing strategies. It may be better presented in a table that presents additional demographic information such as Gender of the children.

Thank you for your suggestion. We agree and Tables 2 and 3 now present yield. 

2. Is additional demographic information such as gender available? This would useful to add to the paper

We agree with you again and have disaggregated by gender in Table 3.

3. Line 196: What happened to the other children that were not enrolled? What was the reason for non-enrolment? Do we know if they were subsequently linked to care as including the non-enrolled in the denominator would affect the linkage to care proportion significantly? 

The reasons for non-enrollment are now given in the Challenges section. We agree with you that the inclusion of this data would add great value and potentially shift the results considerably. Unfortunately, while attempts were made to track them, we were unsuccessful in finding these children.

4. Line 204: why are there no results shown? As noted above this result is critical in understanding weather or not the e-voucher impacted linkage to care.

We have revised this analysis and now show the results in Figure 2. 

Discussion

1. Line 248: the timely client follow-up support is not previously defined? How was this different from facility care?

Great point—we went back and defined linkage to care to be within 3 months of their HIV-positive test date (line 206). We also have re-worded ‘timely’ to be ‘active’.

2. Line 247: it will be best if the proportion of children lost be included in the results as mentioned above. How was “lost” defined? Is this that there was no record of them at the clinic when you followed up? If so that may be better categorised as “not linked to care”? Again, the follow up procedure between diagnosis to linkage should be clearly described in the paper.\\

We have added in the total number of HIV-positive children identified and the proportion enrolled in lines 263-264 of the Results. In the Discussion we have also clarified that “lost” means they were not enrolled into the project’s mobile application and therefore also excluded from the follow-up cohort.

---

## [Decision Letter · Decision Letter 1]

29 Jan 2020

PONE-D-19-26812R1

Beyond the Facility: An Evaluation of Seven Community-Based Pediatric HIV Testing Strategies and Linkage to Care Outcomes in a High Prevalence, Resource-Limited Setting

PLOS ONE

Dear Ms Sindelar,

Thank you for submitting your manuscript to PLOS ONE. After careful consideration, we feel that it has merit but does not fully meet PLOS ONE’s publication criteria as it currently stands. Therefore, we invite you to submit a revised version of the manuscript that addresses the points raised during the review process.

We would appreciate receiving your revised manuscript by Mar 14 2020 11:59PM. To enhance the reproducibility of your results, we recommend that if applicable you deposit your laboratory protocols in protocols.io, where a protocol can be assigned its own identifier (DOI) such that it can be cited independently in the future. For instructions see: http://journals.plos.org/plosone/s/submission-guidelines#loc-laboratory-protocols

We look forward to receiving your revised manuscript.

Kind regards,

Marcel Yotebieng, M.D., MPH, Ph.D

Academic Editor

PLOS ONE

Additional Editor Comments (if provided):

We have heard back from reviewers. though they all appreciate the effort that was put in the revision, there are still major issues to clarify.

Regarding reviewer #3 comment about the results of HIV testing of the mother, I agree with him/her that "Omitting or not reporting on the status of the mother limits the generalizability and usefulness of the results presented". Please if where ever the HIV status of the mother is known, report it. otherwise address this as a limitation in the discussion

I also agree with reviewer #2 on the need to check the consistency of the results including the negative binomial model.

Reviewers' comments:

Reviewer's Responses to Questions

**Comments to the Author**

1. If the authors have adequately addressed your comments raised in a previous round of review and you feel that this manuscript is now acceptable for publication, you may indicate that here to bypass the “Comments to the Author” section, enter your conflict of interest statement in the “Confidential to Editor” section, and submit your "Accept" recommendation.

Reviewer #1: (No Response)

Reviewer #2: (No Response)

Reviewer #3: All comments have been addressed

2. Is the manuscript technically sound, and do the data support the conclusions?

Reviewer #1: Partly

Reviewer #2: Partly

Reviewer #3: Yes

3. Has the statistical analysis been performed appropriately and rigorously? 

Reviewer #1: Yes

Reviewer #2: No

Reviewer #3: Yes

4. Have the authors made all data underlying the findings in their manuscript fully available?

Reviewer #1: Yes

Reviewer #2: Yes

Reviewer #3: Yes

5. Is the manuscript presented in an intelligible fashion and written in standard English?

Reviewer #1: Yes

Reviewer #2: Yes

Reviewer #3: (No Response)

6. Review Comments to the Author

Reviewer #1: Thank you for submitting this revised paper. There is important information being presented and I think that with revision, this paper may be suitable for publication. However, I feel there is one central design flaw that still needs to be addressed. It is unclear to me why the authors of this study would not have ascertained and then reported on the status of the mother of the children being tested through these community based strategies. For PITC, when children are sick and in a health facility, it makes sense for both clinical and practical reasons for the children to be tested directly, regardless of the status of the parent. This is not the case for the testing of healthy children in a community based setting. Testing children of HIV-negative mothers, as seems likely to have been done in this program, is not efficient, as the results of this study clearly show. Testing all children is appropriate for a population based survey like the PHIAs. But that is not what is being reported on here. This paper is looking at practical strategies that countries can implement for community based case finding of children. Omitting or not reporting on the status of the mother limits the generalizability and usefulness of the results presented. Unless this is addressed clearly, I cannot recommend this paper for publication.

Reviewer #2: Summary of revised paper

This article from Sindelar et al has been substantially and meaningfully revised to address reviewer comments. Analyses have been redone and presented in clearer ways and more context has been given. Despite all of these well-conducted revisions, some major internal inconsistencies remain in this paper that would need to be addressed for it to be scientifically accurate. Additionally, there are several cases where easily identifiable and very relevant literature was not able to be found by the authors, which diminishes the sense that a careful detailed eye was applied to other areas of revision within the paper.

Major essential comments

1. Table 2 does not include the number of children who tested positive, which makes it impossible to check many of the subsequent analyses. Below are specific items to address to allow the paper to be accurately evaluated:

a. The authors’ revised “Objectives” section is great. However, the columns in Table 2 do not match their objective, which states that comparison will be done between # tested, # tested positive, and proportion tested positive (yield). The table shows a new proportion that was not previously described and is not well interpreted in the text, and fails to include the number who tested positive in each of the categories. Suggest adding this column. The authors suggest this was omitted because of word count or space restrictions; in this case, I’d suggest removing the column showing the proportion in the column to the left of yield.

b. Related to the previous comment, when I tried to back calculate the number of positives in each of the strata (as suggested by the authors in their response letter), the presented yields did not make sense. For example, the stated yield for D2D Index testing is 0.66% with a denominator of 229 children; if 1 child had tested positive, the yield would have been 0.437%; if 2 had tested positive, it would have been 0.873%; neither of these is the reported 0.66%. I checked several numbers across this table and all suggested either incorrect math or rounding errors. Suggest adding the column with the number of people who tested positive and then rechecking all proportions listed in the paper.

2. It is not possible to fully evaluate the accuracy of the negative binomial regression that the authors present. Despite the changes made in this revision, I still have concerns about the accuracy of this analysis. While the authors tried to address this concern in the revision by noting for D2D Index that 4.95 translates to 0.50, which is close to the 0.66% presented, that logic does NOT hold for the Targeted (2.23 vs 0.63%) and the Facility Index testing (1.34 vs 0.66%). Additionally, if this model was used to adjust for differences in time spent using each approach, we would not expect these numbers to be comparable. Overall, this statistical analysis cannot be evaluated for accuracy and appropriateness until the number of positive individuals is included in Table 2 and the proportions of yield are revised to be accurate.

3. The authors suggest that despite extensive literature review, there were no studies identified that addressed the impact of incentives on linkage to care. I’ve provided a number of articles that were identified using the search terms, “incentive linkage to care HIV” in PubMed. Even if these are not perfectly aligned to what the authors were searching for, it is inappropriate to present the results of this incentive analysis without contextualizing it in any of the incentives literature:

A Conditional Economic Incentive Fails to Improve Linkage to Care and Antiretroviral Therapy Initiation Among HIV-Positive Adults in Cape Town, South Africa.

Maughan-Brown B, Smith P, Kuo C, Harrison A, Lurie MN, Bekker LG, Galárraga O.

AIDS Patient Care STDS. 2018 Feb;32(2):70-78. doi: 10.1089/apc.2017.0238.

Financial Incentives for Linkage to Care and Viral Suppression Among HIV-Positive Patients: A Randomized Clinical Trial (HPTN 065).

El-Sadr WM, Donnell D, Beauchamp G, Hall HI, Torian LV, Zingman B, Lum G, Kharfen M, Elion R, Leider J, Gordin FM, Elharrar V, Burns D, Zerbe A, Gamble T, Branson B; HPTN 065 Study Team.

JAMA Intern Med. 2017 Aug 1;177(8):1083-1092. doi: 10.1001/jamainternmed.2017.2158.

Economic strengthening for HIV testing and linkage to care: a review of the evidence.

Swann M.

AIDS Care. 2018;30(sup3):85-98. doi: 10.1080/09540121.2018.1476665. Review

https://www.ncbi.nlm.nih.gov/pmc/articles/PMC4699403/

Economic strengthening for HIV testing and linkage to care: a review of the evidence.

Swann M.

AIDS Care. 2018;30(sup3):85-98. doi: 10.1080/09540121.2018.1476665. Review.

A combination intervention strategy to improve linkage to and retention in HIV care following diagnosis in Mozambique: A cluster-randomized study.

Elul B, Lamb MR, Lahuerta M, Abacassamo F, Ahoua L, Kujawski SA, Tomo M, Jani I.

PLoS Med. 2017 Nov 14;14(11):e1002433. doi: 10.1371/journal.pmed.1002433. eCollection 2017 Nov.

High-Yield HIV Testing, Facilitated Linkage to Care, and Prevention for Female Youth in Kenya (GIRLS Study): Implementation Science Protocol for a Priority Population.

Inwani I, Chhun N, Agot K, Cleland CM, Buttolph J, Thirumurthy H, Kurth AE.

JMIR Res Protoc. 2017 Dec 13;6(12):e179. doi: 10.2196/resprot.8200.

Investigating interventions to increase uptake of HIV testing and linkage into care or prevention for male partners of pregnant women in antenatal clinics in Blantyre, Malawi: study protocol for a cluster randomised trial.

Choko AT, Fielding K, Stallard N, Maheswaran H, Lepine A, Desmond N, Kumwenda MK, Corbett EL.

Trials. 2017 Jul 24;18(1):349. doi: 10.1186/s13063-017-2093-2.

Voucher incentives improve linkage to and retention in care among HIV-infected drug users in Chennai, India.

Solomon SS, Srikrishnan AK, Vasudevan CK, Anand S, Kumar MS, Balakrishnan P, Mehta SH, Solomon S, Lucas GM.

Clin Infect Dis. 2014 Aug 15;59(4):589-95. doi: 10.1093/cid/ciu324. Epub 2014 May 6.

Minor

1. In table 1, D2D Index testing should be relabeled as a targeted strategy. Any strategies that use an index to identify individuals for testing, or apply any other approach for risk stratification among individuals or risk to influence selection of sites is targeted, not blanket testing.

2. The authors clarified that “active” in care is defined as having one or more visits after ART initiation. It is not clear whether this took into consideration the number of visits that one SHOULD have had. For example, a child who had 2 visits after ART initiation, but was expected based on time elapsed to have had 9, should likely be classified as lost to follow up or not retained, but using the authors’ described definition would still be considered active. This approach systematically overestimates retention, which is not scientifically appropriate.

3. The analysis in lines 274-6 should include a statistical comparison and p-value.

4. There is mismatch between Table 2 and Figure 1. Figure 1 still has “Facility-based” as a category, which is not presented in Table 2 and seems strangely high.

5. There is still new data presented in the discussion of this manuscript, which should not happen in a scientific manuscript. The authors state that the data were not presented in the results section because the sample used in the survey was not representative; this is not a reasonable reason for introducing new data in the discussion, unless those data have already been published elsewhere (in which case they should be referenced).

6. The authors were not able to find any papers about linkage to care in the pre or post test and treat era. Here are some papers for consideration from PubMed. Also consider finding the pediatric specific paper from the SEARCH trial. Some include adult populations, but can be included to contextualize:

Predictors of timely linkage-to-ART within universal test and treat in the HPTN 071 (PopART) trial in Zambia and South Africa: findings from a nested case-control study.

Sabapathy K, Mubekapi-Musadaidzwa C, Mulubwa C, Schaap A, Hoddinott G, Stangl A, Floyd S, Ayles H, Fidler S, Hayes R; HPTN 071 (PopART) study team.

J Int AIDS Soc. 2017 Dec;20(4). doi: 10.1002/jia2.25037.

High levels of retention in care with streamlined care and universal test and treat in East Africa.

Brown LB, Havlir DV, Ayieko J, Mwangwa F, Owaraganise A, Kwarisiima D, Jain V, Ruel T, Clark T, Chamie G, Bukusi EA, Cohen CR, Kamya MR, Petersen ML, Charlebois ED; SEARCH Collaboration.

AIDS. 2016 Nov 28;30(18):2855-2864.

Only adults: Understanding the Time Needed to Link to Care and Start ART in Seven HPTN 071 (PopART) Study Communities in Zambia and South Africa.

Seeley J, Bond V, Yang B, Floyd S, MacLeod D, Viljoen L, Phiri M, Simuyaba M, Hoddinott G, Shanaube K, Bwalya C, de Villiers L, Jennings K, Mwanza M, Schaap A, Dunbar R, Sabapathy K, Ayles H, Bock P, Hayes R, Fidler S; HPTN 071 (PopART) study team.

AIDS Behav. 2019 Apr;23(4):929-946. doi: 10.1007/s10461-018-2335-7.

7. The reasons given for no voucher being received are related to distance, which could also be associated with linkage, serving as a confounder. This needs to be mentioned in the limitations section and interepreted.

Reviewer #3: The is a well written paper which provides useful data with potential to influence HTC policy for children and adolescents, an area which is often neglected. The authors have adequately responded to reviewer comments and it is my recommendation that this paper be accepted for publication.

I have no additional comments.

7. PLOS authors have the option to publish the peer review history of their article (what does this mean?). If published, this will include your full peer review and any attached files.

Reviewer #1: No

Reviewer #2: No

Reviewer #3: Yes: Chido Dziva Chikwari

---

## [Author Response · Author response to Decision Letter 1]

14 Mar 2020

Reviewer #1 Comments to the Author

Reviewer #1: Thank you for submitting this revised paper. There is important information being presented and I think that with revision, this paper may be suitable for publication. However, I feel there is one central design flaw that still needs to be addressed. It is unclear to me why the authors of this study would not have ascertained and then reported on the status of the mother of the children being tested through these community based strategies. For PITC, when children are sick and in a health facility, it makes sense for both clinical and practical reasons for the children to be tested directly, regardless of the status of the parent. This is not the case for the testing of healthy children in a community based setting. Testing children of HIV-negative mothers, as seems likely to have been done in this program, is not efficient, as the results of this study clearly show. Testing all children is appropriate for a population based survey like the PHIAs. But that is not what is being reported on here. This paper is looking at practical strategies that countries can implement for community based case finding of children. Omitting or not reporting on the status of the mother limits the generalizability and usefulness of the results presented. Unless this is addressed clearly, I cannot recommend this paper for publication.

We apologize for the lack of clarity and specificity given in our eligibility criteria. Children of HIV-negative mothers were not tested under the M-HIT project, and as such we have revised lines 155-160 to clarify. 

Reviewer #2: Summary of revised paper

This article from Sindelar et al has been substantially and meaningfully revised to address reviewer comments. Analyses have been redone and presented in clearer ways and more context has been given. Despite all of these well-conducted revisions, some major internal inconsistencies remain in this paper that would need to be addressed for it to be scientifically accurate. Additionally, there are several cases where easily identifiable and very relevant literature was not able to be found by the authors, which diminishes the sense that a careful detailed eye was applied to other areas of revision within the paper.

Major essential comments

1. Table 2 does not include the number of children who tested positive, which makes it impossible to check many of the subsequent analyses. Below are specific items to address to allow the paper to be accurately evaluated:

a. The authors’ revised “Objectives” section is great. However, the columns in Table 2 do not match their objective, which states that comparison will be done between # tested, # tested positive, and proportion tested positive (yield). The table shows a new proportion that was not previously described and is not well interpreted in the text, and fails to include the number who tested positive in each of the categories. Suggest adding this column. The authors suggest this was omitted because of word count or space restrictions; in this case, I’d suggest removing the column showing the proportion in the column to the left of yield.

We were worried the table would be too busy and had erroneously thought the reader could calculate the number who tested positive. You’re correct that it is difficult to do this as we are showing averages in the table with numbers that are not whole integers, and because these numbers are small rounding does play a factor. As such, we have revised both Tables 2 and 3 to include the number testing HIV-positive. We left in the proportion tested (of each age group), as it was crucial to the project to determine if these strategies were successful at reaching substantial proportions of children. This was added to Objective 1 to better highlight its importance to the study.

b. Related to the previous comment, when I tried to back calculate the number of positives in each of the strata (as suggested by the authors in their response letter), the presented yields did not make sense. For example, the stated yield for D2D Index testing is 0.66% with a denominator of 229 children; if 1 child had tested positive, the yield would have been 0.437%; if 2 had tested positive, it would have been 0.873%; neither of these is the reported 0.66%. I checked several numbers across this table and all suggested either incorrect math or rounding errors. Suggest adding the column with the number of people who tested positive and then rechecking all proportions listed in the paper.

Thank you for pointing this out to us—you are correct, and we did make corrections to the “All M-HIT” age and sex yields, and as mentioned above add in the numbers testing HIV-positive to Tables 2 and 3.

2. It is not possible to fully evaluate the accuracy of the negative binomial regression that the authors present. Despite the changes made in this revision, I still have concerns about the accuracy of this analysis. While the authors tried to address this concern in the revision by noting for D2D Index that 4.95 translates to 0.50, which is close to the 0.66% presented, that logic does NOT hold for the Targeted (2.23 vs 0.63%) and the Facility Index testing (1.34 vs 0.66%). Additionally, if this model was used to adjust for differences in time spent using each approach, we would not expect these numbers to be comparable. Overall, this statistical analysis cannot be evaluated for accuracy and appropriateness until the number of positive individuals is included in Table 2 and the proportions of yield are revised to be accurate. 

We have added in the number of positive individuals to Tables 2 and 3 and hope that sheds better light on how we modeled the number of HIV positive children for all testing strategies—M-HIT and facility-based testing. In addition, we would be happy to share our Stata do-file if that would help the reviewer feel more comfortable with this analysis; we consulted an epidemiologist and she confirmed the methods were sound. The model was run to compare the seven community-based strategies with facility-based testing. In doing so, we needed to account for the different number of months each strategy was implemented, the different numbers tested in each strategy, and the overlapping spatial dimensions (multiple strategies could have been implemented in the same community councils, a sub-division of Lesotho’s districts). Finally, the example we gave was meant to reconcile a previous concern about the yields being very different in the model—we had only wanted to show that the yields were in fact similar.

3. The authors suggest that despite extensive literature review, there were no studies identified that addressed the impact of incentives on linkage to care. I’ve provided a number of articles that were identified using the search terms, “incentive linkage to care HIV” in PubMed. Even if these are not perfectly aligned to what the authors were searching for, it is inappropriate to present the results of this incentive analysis without contextualizing it in any of the incentives literature:

A Conditional Economic Incentive Fails to Improve Linkage to Care and Antiretroviral Therapy Initiation Among HIV-Positive Adults in Cape Town, South Africa.

Maughan-Brown B, Smith P, Kuo C, Harrison A, Lurie MN, Bekker LG, Galárraga O.

AIDS Patient Care STDS. 2018 Feb;32(2):70-78. doi: 10.1089/apc.2017.0238.

Financial Incentives for Linkage to Care and Viral Suppression Among HIV-Positive Patients: A Randomized Clinical Trial (HPTN 065).

El-Sadr WM, Donnell D, Beauchamp G, Hall HI, Torian LV, Zingman B, Lum G, Kharfen M, Elion R, Leider J, Gordin FM, Elharrar V, Burns D, Zerbe A, Gamble T, Branson B; HPTN 065 Study Team.

JAMA Intern Med. 2017 Aug 1;177(8):1083-1092. doi: 10.1001/jamainternmed.2017.2158.

Economic strengthening for HIV testing and linkage to care: a review of the evidence.

Swann M.

AIDS Care. 2018;30(sup3):85-98. doi: 10.1080/09540121.2018.1476665. Review

https://www.ncbi.nlm.nih.gov/pmc/articles/PMC4699403/

Economic strengthening for HIV testing and linkage to care: a review of the evidence.

Swann M.

AIDS Care. 2018;30(sup3):85-98. doi: 10.1080/09540121.2018.1476665. Review.

A combination intervention strategy to improve linkage to and retention in HIV care following diagnosis in Mozambique: A cluster-randomized study.

Elul B, Lamb MR, Lahuerta M, Abacassamo F, Ahoua L, Kujawski SA, Tomo M, Jani I.

PLoS Med. 2017 Nov 14;14(11):e1002433. doi: 10.1371/journal.pmed.1002433. eCollection 2017 Nov.

High-Yield HIV Testing, Facilitated Linkage to Care, and Prevention for Female Youth in Kenya (GIRLS Study): Implementation Science Protocol for a Priority Population.

Inwani I, Chhun N, Agot K, Cleland CM, Buttolph J, Thirumurthy H, Kurth AE.

JMIR Res Protoc. 2017 Dec 13;6(12):e179. doi: 10.2196/resprot.8200.

Investigating interventions to increase uptake of HIV testing and linkage into care or prevention for male partners of pregnant women in antenatal clinics in Blantyre, Malawi: study protocol for a cluster randomised trial.

Choko AT, Fielding K, Stallard N, Maheswaran H, Lepine A, Desmond N, Kumwenda MK, Corbett EL.

Trials. 2017 Jul 24;18(1):349. doi: 10.1186/s13063-017-2093-2.

Voucher incentives improve linkage to and retention in care among HIV-infected drug users in Chennai, India.

Solomon SS, Srikrishnan AK, Vasudevan CK, Anand S, Kumar MS, Balakrishnan P, Mehta SH, Solomon S, Lucas GM.

Clin Infect Dis. 2014 Aug 15;59(4):589-95. doi: 10.1093/cid/ciu324. Epub 2014 May 6.

Thank you for the tremendous effort you made to provide relevant and valuable research to adequately bulk up the incentives discussion. To clarify, we did not suggest that such papers did not exist (as was stated on lines 371-382 of the second submission), however we felt that because of the inherent difference in conditional and unconditional incentives, it could be misleading and inaccurate to compare their outcomes. We understand your concerns around providing a broader contextualization of these outcomes for the audience and have expanded this discussion with additional literature as suggested. 

Minor

1. In table 1, D2D Index testing should be relabeled as a targeted strategy. Any strategies that use an index to identify individuals for testing, or apply any other approach for risk stratification among individuals or risk to influence selection of sites is targeted, not blanket testing.

Change made in Table 1.

2. The authors clarified that “active” in care is defined as having one or more visits after ART initiation. It is not clear whether this took into consideration the number of visits that one SHOULD have had. For example, a child who had 2 visits after ART initiation, but was expected based on time elapsed to have had 9, should likely be classified as lost to follow up or not retained, but using the authors’ described definition would still be considered active. This approach systematically overestimates retention, which is not scientifically appropriate.

There are several reasons we utilized this terminology. It was previously reported that no standard definition for retention in care exists [1], and therefore we purposefully chose a simplified definition of ‘active’. Although this too could be defined differently, we were somewhat limited by our data. According to Lesotho’s MoH guidelines, all children should have at least one ART visit by 3-months, so for the sake of consistency with their reporting (as was done as much as possible in the M-HIT study) we adopted this definition. Finally, we do not believe that the opposite of active is lost-to-follow-up, nor that active means the same thing as ‘retained’, and again purposefully did not use these terms. 

That being said, we understand your concern around utilizing the word ‘active’ to describe this outcome as it can mislead the reader. In an effort to be as clear as possible and use streamlined terminology, we found wide adoption (including the WHO) of the more inclusive term “engaged in care,” which we believe is a more accurate description of the outcomes being presented in this research [2-5]. 

3. The analysis in lines 274-6 should include a statistical comparison and p-value.

We have conducted a Wilcoxon rank-sum test and added the p-value.

4. There is mismatch between Table 2 and Figure 1. Figure 1 still has “Facility-based” as a category, which is not presented in Table 2 and seems strangely high.

Figure 1 should be compared to Table 3, as both of those show data on the ‘Children’ age-group. We have included facility-based testing in Table 3 where we could. As described above, Figure 1 is modeled data, and the purpose was to include this important comparison to the facility-based testing. We would expect facility-based testing to find the highest number of HIV-positive children as the traditional venue where children are tested, particularly those who are ill and symptomatic. Ideally, we would have included facility-based data in Table 2 as well, however the MoH age stratifications do not provide the breakdown between adults and adolescents. 

5. There is still new data presented in the discussion of this manuscript, which should not happen in a scientific manuscript. The authors state that the data were not presented in the results section because the sample used in the survey was not representative; this is not a reasonable reason for introducing new data in the discussion, unless those data have already been published elsewhere (in which case they should be referenced).

This section has been revised so that it no longer cites results from the survey and instead frames it as a discussion of all potential benefits of the mobile outreach clinics. 

6. The authors were not able to find any papers about linkage to care in the pre or post test and treat era. Here are some papers for consideration from PubMed. Also consider finding the pediatric specific paper from the SEARCH trial. Some include adult populations, but can be included to contextualize:

Predictors of timely linkage-to-ART within universal test and treat in the HPTN 071 (PopART) trial in Zambia and South Africa: findings from a nested case-control study.

Sabapathy K, Mubekapi-Musadaidzwa C, Mulubwa C, Schaap A, Hoddinott G, Stangl A, Floyd S, Ayles H, Fidler S, Hayes R; HPTN 071 (PopART) study team.

J Int AIDS Soc. 2017 Dec;20(4). doi: 10.1002/jia2.25037.

High levels of retention in care with streamlined care and universal test and treat in East Africa.

Brown LB, Havlir DV, Ayieko J, Mwangwa F, Owaraganise A, Kwarisiima D, Jain V, Ruel T, Clark T, Chamie G, Bukusi EA, Cohen CR, Kamya MR, Petersen ML, Charlebois ED; SEARCH Collaboration.

AIDS. 2016 Nov 28;30(18):2855-2864.

Only adults: Understanding the Time Needed to Link to Care and Start ART in Seven HPTN 071 (PopART) Study Communities in Zambia and South Africa.

Seeley J, Bond V, Yang B, Floyd S, MacLeod D, Viljoen L, Phiri M, Simuyaba M, Hoddinott G, Shanaube K, Bwalya C, de Villiers L, Jennings K, Mwanza M, Schaap A, Dunbar R, Sabapathy K, Ayles H, Bock P, Hayes R, Fidler S; HPTN 071 (PopART) study team.

AIDS Behav. 2019 Apr;23(4):929-946. doi: 10.1007/s10461-018-2335-7.

Thank you, we have further contextualized this discussion point as suggested. 

7. The reasons given for no voucher being received are related to distance, which could also be associated with linkage, serving as a confounder. This needs to be mentioned in the limitations section and interpreted.

The reasons we gave for no e-voucher being received were in lines 437-438: “Additionally, some children did not receive an e-voucher due to network failures in remote locations, client errors in M-Pesa registration, or general system errors.” Distance to the health facility was irrelevant as all e-vouchers were sent via mobile phone and the cash could be collected through any m-pesa agent nationwide. The money could also be transferred to a transport operator electronically if no agents were near.

References

1. Rollins N, Essajee S, Bellare N, Doherty M, Hirnschall G. Improving Retention in Care Among Pregnant Women and Mothers Living With HIV. JAIDS Journal of Acquired Immune Deficiency Syndromes. 2017;75:S111-S114.

2. World Health Organization. Global Health Sector Strategy on HIV: 2016–2021. Geneva: World Health Organization; 2016. Available from: https://apps.who.int/iris/bitstream/handle/10665/246178/WHO-HIV-2016.05-eng.pdf

3. Bengtson A, Kumwenda W, Lurie M, Klyn B, Owino M, Miller W et al. Improving Monitoring of Engagement in HIV Care for Women in Option B+: A Pilot Test of Biometric Fingerprint Scanning in Lilongwe, Malawi. AIDS and Behavior. 2019;24(2):551-559.

4. Cheever L. Engaging HIV-Infected Patients in Care: Their Lives Depend on It. Clinical Infectious Diseases. 2007;44(11):1500-1502.

5. Gardner E, McLees M, Steiner J, del Rio C, Burman W. The Spectrum of Engagement in HIV Care and its Relevance to Test-and-Treat Strategies for Prevention of HIV Infection. 2020.

---

## [Decision Letter · Decision Letter 2]

13 May 2020

PONE-D-19-26812R2

Beyond the Facility: An Evaluation of Seven Community-Based Pediatric HIV Testing Strategies and Linkage to Care Outcomes in a High Prevalence, Resource-Limited Setting

PLOS ONE

Dear Ms Sindelar,

Thank you for submitting your manuscript to PLOS ONE. After careful consideration, we feel that it has merit but does not fully meet PLOS ONE’s publication criteria as it currently stands. Therefore, we invite you to submit a revised version of the manuscript that addresses the points raised during the review process.

We would appreciate receiving your revised manuscript by Jun 27 2020 11:59PM. To enhance the reproducibility of your results, we recommend that if applicable you deposit your laboratory protocols in protocols.io, where a protocol can be assigned its own identifier (DOI) such that it can be cited independently in the future. For instructions see: http://journals.plos.org/plosone/s/submission-guidelines#loc-laboratory-protocols

We look forward to receiving your revised manuscript.

Kind regards,

Marcel Yotebieng, M.D., MPH, Ph.D

Academic Editor

PLOS ONE

Additional Editor Comments (if provided):

Sorry for the delay getting back to you on this. I have had a phone discussion with Reviewer #1 about their concern. the yield to the testing is low and it is important for country program and other funding agencies, to clearly understand who was tested. if as stated only “Children of HIV-negative mothers were not tested under the M-HIT project”, it is important to made this clear and maybe separate children tested in this strategy from other blanket testing strategy.

This is a very large testing effort.

Reviewers' comments:

Reviewer's Responses to Questions

**Comments to the Author**

1. If the authors have adequately addressed your comments raised in a previous round of review and you feel that this manuscript is now acceptable for publication, you may indicate that here to bypass the “Comments to the Author” section, enter your conflict of interest statement in the “Confidential to Editor” section, and submit your "Accept" recommendation.

Reviewer #1: (No Response)

Reviewer #2: (No Response)

Reviewer #4: (No Response)

2. Is the manuscript technically sound, and do the data support the conclusions?

Reviewer #1: No

Reviewer #2: Partly

Reviewer #4: Yes

3. Has the statistical analysis been performed appropriately and rigorously? 

Reviewer #1: N/A

Reviewer #2: No

Reviewer #4: Yes

4. Have the authors made all data underlying the findings in their manuscript fully available?

Reviewer #1: Yes

Reviewer #2: Yes

Reviewer #4: No

5. Is the manuscript presented in an intelligible fashion and written in standard English?

Reviewer #1: Yes

Reviewer #2: Yes

Reviewer #4: Yes

6. Review Comments to the Author

Reviewer #1: Thanks for the response to my concern. The authors have stated that “Children of HIV-negative mothers were not tested under the M-HIT project”, which is a really important statement. What it implies is that the M-HIT project had a system for screening for the exposure status of children and then directing their testing based on that screening process. This statement implies that they had a way of determining whether the mother was HIV infected or HIV-negative. The authors need to provide more detail on this as it would be crucial for other countries and programs to understand how they managed this as they design their own community testing programs. The actual data collection form and SOP should be provided. It would also be essential if the authors have it to provide information on how many mothers were screened, how many were infected versus negative, as well as how many children had a deceased mother or a status unknown mother.

I am still concerned about whether in fact all or even a majority of the children were HIV-exposed and eligible for testing and that concern is based on the numbers. According to the text an average of 2433 were done per month over 30 months which would total to 72,990 tests in 2 districts in Lesotho. This number is discrepant to the MHIT Testing Strategies Dataset provided which has a total of 406,983 test (46% in children would give roughly 187,000. I didn’t go through to sort it to get the exact number, but the authors need to double-check this discrepancy). Regardless of which number is correct, both would be very high for a testing strategy that was restricted to HIV-exposed children. To give a sense of expected numbers see exercise below utilizing numbers from Spectrum. According to this exercise, there are only roughly 150,000 exposed children in all of Lesotho. Using the numbers given in the text (72,990), the authors propose that they have tested half this number in just two districts. The yield seen in this study is less than what would be expected if you just tested all children in the general population at random, which is not what the authors have stated was done in this program.

Numbers for Lesotho from Spectrum 2018 and 2019

General Population= 2,049,311

Population <15= 651,743

Estimated <15 exposed to HIV= 151,649 (calculated as mothers needing PMTCT per year x .95x 15 years)

Estimated <15 HIV-infected= 12,134

Prevalence<15= 12,143/651,743=

<15 HIV+ on ART= 8,499

<15 HIV+ not on ART= 3,635

Expected Yield General Population Testing= 3,635/651,743= 0.6%

Expected Yield HIV Exposed Restricted Testing= 3635/151,649= 2.4%

Perhaps, I have made an error in my calculations. I would love to hear back from the authors on this. I must again emphasize the primary importance of describing clearly how the children in this study came to be tested for the downstream interpretation of these results and the generalizability of the findings to other programs and countries.

Reviewer #2: Thank you for making revisions to this paper. All but the following two points were sufficiently addressed in the most recent submission.

The modeled number of positive children analysis is still not clear, even when compared to the newly presented number of children tested. I suggest removing it from the manuscript as it does not add additional insight about how to prioritize testing strategies.

While the evouchers were sent by mobile phone, and therefore getting to the health facility was not a problem in receiving money, the authors still state that "remote locations" was associated with network failures and accounted for some evouchers not being received; "remote locations" is also likely associated with distance to a clinic and therefore likelihood of linkage to care. Given this potential for confounding, a note should be added to the limitations to qualify this.

Reviewer #4: 1) It is unclear how “positive yield” was defined and calculated. Take Table 2 Adults category as an example. In the first row, Tested positive (30.80) / Tested (1182) = 2.60%, which is the positive yield shown in the table. But how the All M-HIT positive yield was calculated, as 17.40 / 2327 = 0.75%, which does not equal to 3.73%.

7. PLOS authors have the option to publish the peer review history of their article (what does this mean?). If published, this will include your full peer review and any attached files.

Reviewer #1: No

Reviewer #2: No

Reviewer #4: No

---

## [Author Response · Author response to Decision Letter 2]

27 Jun 2020

Reviewer #1: Thanks for the response to my concern. The authors have stated that “Children of HIV-negative mothers were not tested under the M-HIT project”, which is a really important statement. What it implies is that the M-HIT project had a system for screening for the exposure status of children and then directing their testing based on that screening process. This statement implies that they had a way of determining whether the mother was HIV infected or HIV-negative. The authors need to provide more detail on this as it would be crucial for other countries and programs to understand how they managed this as they design their own community testing programs. The actual data collection form and SOP should be provided. It would also be essential if the authors have it to provide information on how many mothers were screened, how many were infected versus negative, as well as how many children had a deceased mother or a status unknown mother.

I am still concerned about whether in fact all or even a majority of the children were HIV-exposed and eligible for testing and that concern is based on the numbers. According to the text an average of 2433 were done per month over 30 months which would total to 72,990 tests in 2 districts in Lesotho. This number is discrepant to the MHIT Testing Strategies Dataset provided which has a total of 406,983 test (46% in children would give roughly 187,000. I didn’t go through to sort it to get the exact number, but the authors need to double-check this discrepancy). Regardless of which number is correct, both would be very high for a testing strategy that was restricted to HIV-exposed children. To give a sense of expected numbers see exercise below utilizing numbers from Spectrum. According to this exercise, there are only roughly 150,000 exposed children in all of Lesotho. Using the numbers given in the text (72,990), the authors propose that they have tested half this number in just two districts. The yield seen in this study is less than what would be expected if you just tested all children in the general population at random, which is not what the authors have stated was done in this program.

Numbers for Lesotho from Spectrum 2018 and 2019

General Population= 2,049,311

Population <15= 651,743

Estimated <15 exposed to HIV= 151,649 (calculated as mothers needing PMTCT per year x .95x 15 years)

Estimated <15 HIV-infected= 12,134

Prevalence<15= 12,143/651,743=

<15 HIV+ on ART= 8,499

<15 HIV+ not on ART= 3,635

Expected Yield General Population Testing= 3,635/651,743= 0.6%

Expected Yield HIV Exposed Restricted Testing= 3635/151,649= 2.4%

Perhaps, I have made an error in my calculations. I would love to hear back from the authors on this. I must again emphasize the primary importance of describing clearly how the children in this study came to be tested for the downstream interpretation of these results and the generalizability of the findings to other programs and countries.

Author Response

First, we really want to thank you for your thorough examination of our research and the additional effort you put forth in understanding its context. It is also our primarily objective to inform future programs, and we are grateful for the clarity your questions are bringing this paper.

We misconstrued the eligibility terms for children (>18 months) by conveying that it included only those with a mother who is positive or has an unknown status. This statement is only true for infants (<18 months). The majority of the children (>18 months) tested through M-HIT were not exposed; however, they were eligible per the Ministry of Health guidelines. The guidelines state that all individuals with an unknown status are eligible for testing. For those older than 18 months, all were offered testing, regardless of maternal status and exposure history. There is one algorithm that is inclusive of all children >18 months, adolescents, and adults with additional conditions for adolescents and adults as already outlined in the paper. Lesotho has always embraced an inclusive ‘Know Your Status’ philosophy for testing and we apologize for our error in communicating that critical piece of information. 

At the time of M-HIT implementation, additional risk-based stratification was only used for re-testing protocols to determine frequency, not for initial testing. When determining eligibility, healthcare workers examined the child’s health book to identify whether their status was known or unknown. 

It is important to note that both index strategies did not follow the MOH eligibility requirements. In the index strategies, all household members were offered HTS as these individuals were thought to be at higher risk of infection. The manuscript has been updated to reflect all of these HTS eligibility requirements.

We have also provided all MoH HTS algorithms in Annex 1 – 3 of this document (0-9 months, 9-18 months, and children/adolescents/adults). These were taken directly from the Lesotho 2013 MOH Guidelines. This full document will also be uploaded as part of this submission process for your reference. 

Reviewer #2: Thank you for making revisions to this paper. All but the following two points were sufficiently addressed in the most recent submission.

The modeled number of positive children analysis is still not clear, even when compared to the newly presented number of children tested. I suggest removing it from the manuscript as it does not add additional insight about how to prioritize testing strategies.

Author Response

We have removed, per your suggestion.

While the evouchers were sent by mobile phone, and therefore getting to the health facility was not a problem in receiving money, the authors still state that "remote locations" was associated with network failures and accounted for some evouchers not being received; "remote locations" is also likely associated with distance to a clinic and therefore likelihood of linkage to care. Given this potential for confounding, a note should be added to the limitations to qualify this.

Thank you for clarifying this further; we see your point now and have added in a sentence in line 400 – 402.

Reviewer #4: 1) It is unclear how “positive yield” was defined and calculated. Take Table 2 Adults category as an example. In the first row, Tested positive (30.80) / Tested (1182) = 2.60%, which is the positive yield shown in the table. But how the All M-HIT positive yield was calculated, as 17.40 / 2327 = 0.75%, which does not equal to 3.73%.

Author Response

Thank you for catching this error: while for the total number tested I was taking the sum of each strategy, for the total number positive I was taking the overall average—I have revised the table so that both of these numbers are now the overall average for all strategies, for the average number of months each strategy was implemented for (the number of months each strategy was implemented for is in Table 1). The yields are now reflective of positive/tested. For Adults, you will now see 121.82/3265=3.73%. Note that I have updated the “All M-HIT” row in Table 3 to align to the same methods as Table 2.

---

## [Decision Letter · Decision Letter 3]

20 Jul 2020

Beyond the Facility: An Evaluation of Seven Community-Based Pediatric HIV Testing Strategies and Linkage to Care Outcomes in a High Prevalence, Resource-Limited Setting

PONE-D-19-26812R3

Dear Dr. Sindelar,

We’re pleased to inform you that your manuscript has been judged scientifically suitable for publication and will be formally accepted for publication once it meets all outstanding technical requirements.

Kind regards,

Marcel Yotebieng, M.D., MPH, Ph.D

Academic Editor

PLOS ONE

Additional Editor Comments (optional):

Reviewers' comments:

Reviewer's Responses to Questions

**Comments to the Author**

1. If the authors have adequately addressed your comments raised in a previous round of review and you feel that this manuscript is now acceptable for publication, you may indicate that here to bypass the “Comments to the Author” section, enter your conflict of interest statement in the “Confidential to Editor” section, and submit your "Accept" recommendation.

Reviewer #1: (No Response)

Reviewer #2: All comments have been addressed

Reviewer #4: All comments have been addressed

2. Is the manuscript technically sound, and do the data support the conclusions?

Reviewer #1: Partly

Reviewer #2: Yes

Reviewer #4: Yes

3. Has the statistical analysis been performed appropriately and rigorously? 

Reviewer #1: Yes

Reviewer #2: Yes

Reviewer #4: Yes

4. Have the authors made all data underlying the findings in their manuscript fully available?

Reviewer #1: Yes

Reviewer #2: Yes

Reviewer #4: No

5. Is the manuscript presented in an intelligible fashion and written in standard English?

Reviewer #1: Yes

Reviewer #2: Yes

Reviewer #4: Yes

6. Review Comments to the Author

Reviewer #1: Thank you to the authors for clarifying the eligibility criteria. I think that was really important. A couple of more comments:

1) Can you please double-check the total number of tests? According to the text an average of 2433 were done per month over 30 months which would total to 72,990 tests in 2 districts in Lesotho. This number is discrepant to the MHIT Testing Strategies Dataset provided which has a total of 406,983 test (46% in children would give roughly 187,000).

2) I think it would be useful in the conclusion to mention that all children were tested regardless of exposure status per the Know Your Status Policy. What the authors have shown is that there are clearly missed children that can be identified in these community based settings. But what policymakers need to know now is if there can be a more efficient process of identifying them than what was presented in this paper. To make this paper more relevant, I would encourage the authors to at least present that question. There are several groups working on this. In Zim, there has been symptom based screening efforts. What I think will be more effective is using exposure status- i.e. only test kids if mother is positive, deceased, or unavailable. This can be assessed quickly and will dramatically improve yield. CHAI from Malawi presented on such an approach at AIDS 2020. I am sure the authors also from CHAI are aware of this work and it might be good to reference it.

PEE1431 - "Right under our nose": A simple screening tool to identify HIV-positive children outside of the PMTCT program at outpatient departments in Malawi

Speaker

Anna Tallmadge, Clinton Health Access Initiative

Session Name

E-posters Track E

Room

Poster Channel - Track E

Reviewer #2: (No Response)

Reviewer #4: My comments were addressed and I have no further comment.

My comments were addressed and I have no further comment.

My comments were addressed and I have no further comment.

My comments were addressed and I have no further comment.

7. PLOS authors have the option to publish the peer review history of their article (what does this mean?). If published, this will include your full peer review and any attached files.

Reviewer #1: No

Reviewer #2: No

Reviewer #4: No

---

## [Editor Report · Acceptance letter]

3 Aug 2020

PONE-D-19-26812R3 

Beyond the Facility: An Evaluation of Seven Community-Based Pediatric HIV Testing Strategies and Linkage to Care Outcomes in a High Prevalence, Resource-Limited Setting 

Dear Dr. Sindelar:

I'm pleased to inform you that your manuscript has been deemed suitable for publication in PLOS ONE. Congratulations! Your manuscript is now with our production department. 

Kind regards, 

on behalf of

Dr. Marcel Yotebieng 

Academic Editor

PLOS ONE